# A direct observation of up-converted room-temperature phosphorescence in an anti-Kasha dopant-matrix system

Jiuyang Li[1], Xun Li[1], Guangming Wang[1], Xuepu Wang[1], Minjian Wu[1], Jiahui Liu[1] & Kaka Zhang [1] ✉

It is common sense that emission maxima of phosphorescence spectra ($\lambda_P$) are longer than those of fluorescence spectra ($\lambda_F$). Here we report a serendipitous finding of up-converted room-temperature phosphorescence (RTP) with $\lambda_P < \lambda_F$ and phosphorescence lifetime > 0.1 s upon doping benzophenone-containing difluoroboron β-diketonate ($BPBF_2$) into phenyl benzoate matrices. The up-converted RTP is originated from $BPBF_2$'s $T_n$ ($n \geq 2$) states which show typical $^3$n-π* characters from benzophenone moieties. Detailed studies reveal that, upon intersystem crossing from $BPBF_2$'s $S_1$ states of charge transfer characters, the resultant $T_1$ and $T_n$ states build $T_1$-to-$T_n$ equilibrium. Because of their $^3$n-π* characters, the $T_n$ states possess large phosphorescence rates that can strongly compete RTP($T_1$) to directly emit RTP($T_n$) which violates Kasha's rule. The direct observation of up-converted RTP provides deep understanding of triplet excited state dynamics and opens an intriguing pathway to devise visible-light-excitable deep-blue afterglow emitters, as well as stimuli-responsive afterglow materials.

Manipulation of excited states represents a central topic in the fields of photofunctional materials. A deep understanding of triplet excited state property is of vital importance for devising high-performance room-temperature phosphorescence (RTP) and other luminescent materials[1-10]. Besides the manipulation of $T_1$ states, control of the photophysical behaviors of higher triplet excited states ($T_n$, $n \geq 2$) is also very important because $T_n$ states can mediate $S_1$ to $T_1$ intersystem crossing (ISC) in organic RTP systems and serve as candidates for fabrication of RTP materials with intriguing properties[11-17]. For example, it is well-known that $T_2$ states of $^3$π-π* characters in benzophenone systems can strongly facilitate ISC from $^1$n-π* ($S_1$ state) to $^3$n-π* ($T_1$ state)[2,12]. Benzophenone systems possess very high $\Phi_{ISC}$ but relatively short phosphorescence lifetimes on the order of 0.1 to 1.0 ms; $T_1$ state of $^3$n-π* characters has large phosphorescence decay rate constants ($k_P$). Recent studies show the achievement of both high RTP quantum yields ($\Phi_P$) and relatively long phosphorescence lifetimes ($\tau_P$) by adjusting n-π* and π-π* compositions in $S_1$ and $T_1$ states, as well as in

$T_n$ ($n \geq 2$) states[13-15]. Molecular aggregation has also been demonstrated to control the properties of $T_n$ ($n \geq 2$) states[11,18-20]. Upon aggregation, the electronic interactions between chromophores can cause energy splitting to give rise to multiple close-lying $T_n$ states[19,20]. As a result, more $S_1$-$T_n$ channels with relatively large spin-orbit coupling matrix elements (SOCME) and small $\Delta E_{ST}$ can be generated to enhance ISC processes.

Despite of these essential roles of $T_n$ ($n \geq 2$) states to facilitate ISC, the studies, and understanding on $T_n$ states are mostly restricted to computational studies and ultrafast spectroscopy[21-23]. Photophysical behaviors of $T_n$ ($n \geq 2$) states remain rarely observed and reported in conventional experimental conditions because $T_n$-$T_1$ internal conversion is usually much faster than $T_n$-$S_0$ phosphorescence decay according to Kasha's rule[24,25]. We reason that a direct observation of RTP($T_n$) (RTP from $T_n$ state, $n \geq 2$) would be very important from at least three aspects. First, the observation of RTP($T_n$) by conventional experimental setups and even human eyes can give a straightforward

---

[1]Key Laboratory of Synthetic and Self-Assembly Chemistry for Organic Functional Molecules, Shanghai Institute of Organic Chemistry, University of Chinese Academy of Sciences, Chinese Academy of Sciences, 345 Lingling Road, Shanghai 200032, People's Republic of China. ✉e-mail: zhangkaka@sioc.ac.cn

understanding on excited state dynamics of $T_n$ ($n \geq 2$) states, including their population, conversion, and decay. Second, $RTP(T_n)$ exhibits smaller Stokes shift than $RTP(T_1)$, which would be useful for the fabrication of visible-light-excitable deep-blue RTP materials. To be fair, luminescent materials with large Stokes shift can minimize the interference of scattered light from excitation source; this is an advantage of conventional RTP materials. However, in the case of deep-blue RTP materials, large Stokes shift means that high-energy UV sources (which may destabilize organic materials) are required to excite the materials; for instance, in the reported studies, UV lights of short wavelengths such as 310 nm, 280 nm or even shorter are used to switch on the deep-blue RTP property[5,9]. $RTP(T_n)$ with small Stokes shift would provide a pathway to achieve deep-blue RTP materials that can be excited by visible light or UVA light. Because of the long-lived excited state nature of RTP materials, the interference from excitation source and background fluorescence can be eliminated by time-gated or afterglow mode. Third, the involvement of $RTP(T_n)$ would endow organic systems with $RTP(T_1)$ plus $RTP(T_n)$ dual phosphorescence property. Given that $RTP(T_n)$ and $RTP(T_1)$ possess different population mechanisms and very different phosphorescence decay rates, if some specific stimuli have different influence on $RTP(T_n)$ and $RTP(T_1)$ emission intensities, the organic systems would give significant $RTP(T_n)/RTP(T_1)$ ratiometric response to function as stimuli-responsive RTP materials.

There are very limited examples of the experimental observations of $T_n$-$S_0$ ($n \geq 2$) phosphorescence in conventional conditions[26,27]. In one circumstance, when the $T_2$ states possess typical $^3$n-π* characters, the $k_P$ values of $T_2$-$S_0$ transition can be increased to a large extent to counterbalance the small population of $T_2$ states, leading to $T_2$-$S_0$ phosphorescence[28–30]. The $T_2$-$S_0$ phosphorescence in the dealyed emission spectra has been found to be much weaker than $T_1$-$S_0$ phosphorescence in the reported studies; it is challenging to achieve a major $T_2$-$S_0$ phosphorescence band in the delayed emission spectra due to the fast $T_2$-$T_1$ internal conversion. In another circumstance, the $T_2$ and $T_1$ states have small energy gaps so that $T_2$ and $T_1$ states are in fast equilibrium to exhibit dual phosphorescence behaviors[26,31,32]; the $T_2$-$S_0$ and $T_1$-$S_0$ phosphorescence bands showed large overlap and cannot be resolved very clearly. In the reported studies of both circumstances, the $T_2$ energy levels are lower than $S_1$ states as revealed by steady-state and delayed emission spectra[26,28–32]. In contrast, in computational studies, one may frequently find up-converted $S_1$-$T_n$ ($n \geq 2$) transitions and $T_1$-$T_n$ reverse internal conversion to open forward and reverse ISC channels[11,15,33–35]; such up-converted processes may be not easy to be understood by non-experts because it seems to be thermodynamically unfavorable. Therefore, a direct observation of up-converted $RTP(T_n)$ with $\lambda_P(T_n) < \lambda_F(S_1)$ would have significant impact on the straightforward understanding of the behaviors of higher triplet excited states and the up-converted photophysical processes. However, to the best of our knowledge, in conventional conditions, such $RTP(T_n)$ with $\lambda_P(T_n) < \lambda_F(S_1)$ have been rarely observed by experimental studies; in a reported study[36], $RTP(T_n)$ signals with higher energy levels than $S_1$ states were collected by spectroscopic methods but the $RTP(T_n)$ signals showed short $\tau_P < 10$ ms and cannot be observed by human eyes upon ceasing excitation source.

Here we report a serendipitous finding of up-converted RTP with $\lambda_P < \lambda_F$ and $\tau_P > 0.1$ s upon doping benzophenone-containing difluoroboron β-diketonate ($BPBF_2$) into phenyl benzoate (PhB) matrices. The $BPBF_2$-PhB materials are prepared by rational material design based on dopant-matrix strategy, while the up-converted RTP is from an unexpected observation. The up-converted RTP has been found to originate from $T_n$ ($n \geq 2$) states of $BPBF_2$ which show typical $^3$n-π* characters from benzophenone functional groups. Experimental and computational studies show that the $BPBF_2$-PhB systems have a strong tendency to undergo intersystem crossing. Upon intersystem crossing from $BPBF_2$'s $S_1$ states of charge transfer ($^1$CT) characters, the formed $T_n$ and $T_1$ states build $T_n$-$T_1$ equilibrium via forward and reverse internal

conversion. The $T_n$ states of $^3$n-π* characters possess large $k_P$ values that can strongly compete $RTP(T_1)$ to directly emit $RTP(T_n)$ which violates Kasha's rule.

## Results

### Material fabrication and photophysical measurements

The original purpose of the present study is to fabricate efficient RTP materials in dopant-matrix systems. Since benzophenone systems exhibit strong tendency of intersystem crossing, we synthesized benzophenone-containing difluoroboron β-diketonate compound **1** to serve as luminescent dopants (Fig. 1a). Compound **1** received thorough structural characterization (See Supplementary Information) and photophysical measurements (Supplementary Fig. 1 and Supplementary Table 1). Unlike the reported benzophenone derivatives[12], compound **1** show insignificant room-temperature organic afterglow in its crystal states (Supplementary Fig. 2).

We use dopant-matrix design strategy to construct organic afterglow materials, where the selection of organic matrix is very important. The selection guideline of organic matrix is based on its role in $BF_2$bdk-matrix afterglow system[37], where $BF_2$bdk represents difluoroboron β-diketonate compound. (a) Organic matrix should suppress nonradiative decay and oxygen quenching of $BF_2$bdk's $T_1$ states, so that crystalline matrix is preferred. (b) In $BF_2$bdk-matrix system, organic matrices with carbonyl or ester groups interact with $BF_2$bdk's $S_1$ states via dipole-dipole interactions, lower $BF_2$bdk's $S_1$ levels ($BF_2$bdk's $T_1$ levels are less influenced by matrix's environment), and thus reduce $\Delta E_{ST}$ and facilitate intersystem crossing[38]. This dipole effect in enhancing intersystem crossing has also been proved in a recent reported study[39]. Here phenyl benzoate (PhB) and benzophenone (BP) are used to accommodate $BPBF_2$ because of their crystalline natures and relatively large dipole moments in the ground states; PhB and BP are two of the most frequently used matrices developed in our lab. By doping 0.1 wt% $BPBF_2$ into BP (BP has ground-state dipole moments of 2.96 D as estimated by TD-B3LYP/6-31G(d,p)), the resultant dopant-matrix samples have been found to show insignificant afterglow at room temperature (Supplementary Fig. 3); BP matrix has relatively low $T_1$ level (2.76 eV, estimated from phosphorescence maxima) to receive excited state energy from $BPBF_2$'s $T_1$ states, causing the quenching of organic afterglow in $BPBF_2$-BP samples[40,41]. Cyclo olefin polymer (COP) with high $T_1$ level but insignificant dipole moment has also been test as organic matrix. The $BPBF_2$-COP samples show insignificant room-temperature afterglow (Supplementary Fig. 4).

PhB has ground-state dipole moments of 1.94 D and a high $T_1$ level (3.53 eV and 3.46 eV as calculated by TD-B3LYP/6-31G(d,p) and TD-B3LYP/def2-TZVP(-f), respectively). Upon doping 0.1 wt% **1** into PhB matrices, the obtained **1**-PhB-0.1% materials show blue emission under 365 nm UV lamp and exhibit green afterglow with duration of 3 s after switching off UV lamp. Their steady-state emission spectra show 400-600 nm emission bands with $\lambda_F$ of 437 nm (Fig. 1c). The delayed emission spectra (1 ms delay) show phosphorescence bands ranging from 450-600 nm with $\lambda_P$ of 483 nm and $\tau_P$ of 329 ms, as well as a relatively weak delayed emission band in the higher-energy region with emission lifetime of 303 ms. Surprisingly, this weak delayed emission band has a maximum at 421 nm, which is shorter than the fluorescence maximum ($\lambda_F = 437$ nm). These results are well reproducible (Supplementary Fig. 5). PhB matrices show insignificant phosphorescence upon 365 nm excitation and don't contribute to the 421 nm delayed emission band (Supplementary Fig. 6). Compound **1** was carefully purified by column chromatography followed by recrystallization in spectroscopic grade n-hexane/dichloromethane for three times. Its high purity was confirmed by HPLC measurement (Supplementary Fig. 7). This can rule out the possibility that the higher-energy delayed emission band of **1**-PhB materials originates from some impurity[17]. In our previous studies[34,35], the wavelength of delayed emission maxima

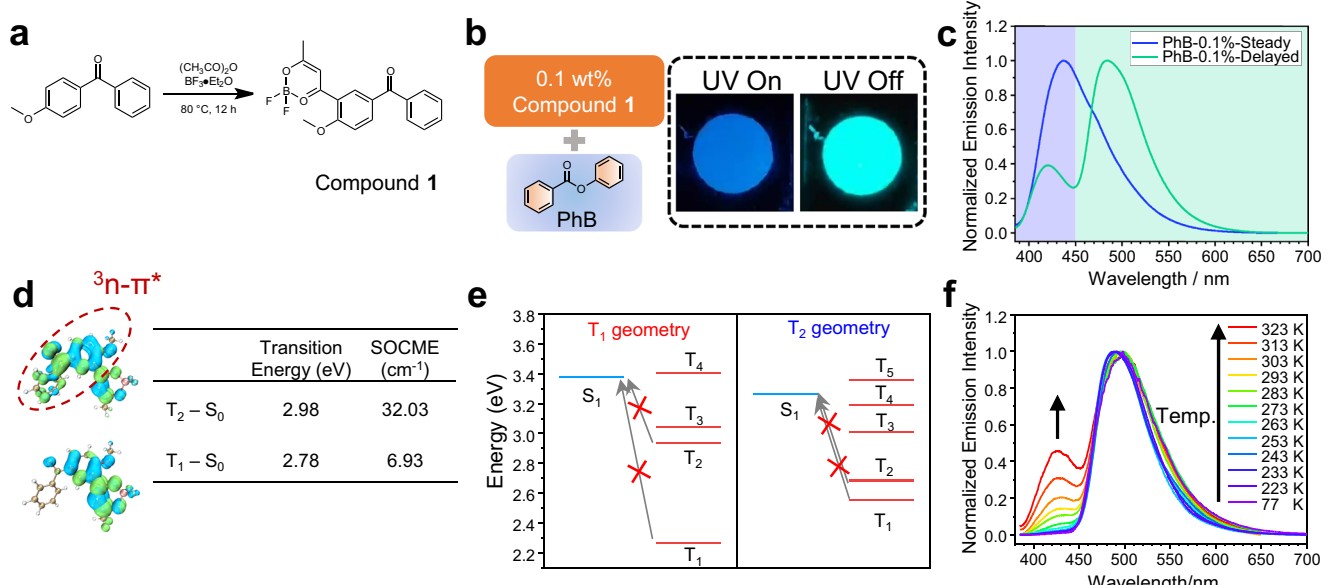

**Fig. 1 | Photophysical properties of compound 1 system. a** Cascade reaction for the synthesis of compound **1**; **b** Illustration of the material composition and photographs of **1**-PhB-0.1% under a 365 UV lamp and after removal of the UV lamp; **c** Normalized steady (blue line) and delayed (green line, 1 ms delay) emission spectra for **1**-PhB-0.1% powder at room temperature; **d** TD-DFT calculation results of T$_1$ and T$_2$ states of compound **1** based on optimized S$_0$ geometry: the isosurface maps of electron-hole density difference (blue and green isosurfaces correspond to hole and electron distribution, respectively), transition energy values and spin-orbital coupling matrix element (SOCME). TD-DFT calculations were performed on ORCA 5.0.3 program with B3LYP/G functional and def2-TZVP(-f) basis set; **e** Calculated excited state energy levels at the optimized geometry of T$_1$ and T$_2$ states of compound **1** with B3LYP/G functional and def2-TZVP(-f) basis set; **f** Temperature-dependent delayed emission spectra (1 ms delay) of **1**-PhB-0.1% powder from 77 K to 323 K.

of TADF-type organic afterglow may be slightly shorter than those in the steady-state emission spectra, which is caused by the aggregation of luminescent dopants; the aggregates cause red shift of steady-state emission spectra but have a less contribution to TADF-type afterglow than monomeric dopants. However, this is not the case in the present study. When the doping concentration is reduced to 0.01% to eliminate the aggregation of luminescent dopants, the higher-energy delayed emission band in the range from 400 to 450 nm with maxima shorter than fluorescence band still exist (Supplementary Fig. 5). TD-DFT calculations of excited state energy levels at the optimized geometry of T$_1$ and T$_2$ states show that the S$_1$ level of compound **1** is much higher than both T$_1$ and T$_2$ states (Fig. 1e). Given that reverse ISC starts from triplet excited states, these results suggest that reverse ISC and TADF would be insignificant in **1**-PhB system. More discussion to rule out the possibility that the 421 nm delayed emission originate from TADF is attached in Supplementary Discussion. In the reported studies[42], the axial and equatorial conformation of the T$_1$ state of phenothiazine-containing compound at room temperature have been found to exhibit higher-energy (local minimum) and lower-energy (global minimum) phosphorescence bands, respectively. At low temperature, the lower-energy bands decrease while the higher-energy bands still exist[42]. One may reason that the 421 nm and 483 nm bands in the present study originate from local minimum and global minimum of **1**'s T$_1$ states, respectively. If this is true, the 421 nm higher-energy band should still exist at low temperature, but here the variable temperature delayed emission spectra of **1**-PhB materials show the decrease and absence of 421 nm band upon lowering temperature (Fig. 1f). Therefore, the conformation-dependent or twist-induced T$_1$ level change is not likely to be the case in the present system with dual RTP property. Besides, in a very recent study of BF$_2$bdk-matrix system reported by our group[43], the twisted BF$_2$bdk compound showed RTP spectral shift upon conformation change, whereas dual RTP has not been observed. In addition, compound **1** has only one conformation in single crystal structure (Supplementary Fig. 40 and Supplementary Data 1). Moreover, TD-DFT calculations have also been performed to investigate the

dependence of T$_1$ levels on **1**'s conformation (Supplementary Fig. 8); the conformation is defined by the twisted angle between aromatic donor and dioxaborine acceptor. The excitation energy of **1**'s T$_1$ state as a function of the twisted angle shows only one energy minimum (Supplementary Fig. 8). These results and analyses exclude the possibility of twist-induced dual RTP in the present system. We realize that here the higher-energy band at 421 nm may originate from T$_n$ ($n \geq 2$) state of $^3$n-π* characters because of the involvement of benzophenone functional groups. This receives support from TD-DFT calculation (Supplementary Fig. 9). Figure 1d shows the isosurface maps of electron-hole density difference between triplet excited states and ground states of compound **1**. It is found that T$_2$ states show typical $^3$n-π* character localized on benzophenone moiety (Fig. 1d). The spin-orbital coupling matrix element (SOCME) value of T$_2$ to S$_0$ transition has been calculated to be as large as 32.03 cm$^{-1}$. With such a large SOCME, it is understandable that the phosphorescence decay from this T$_2$ state should be fast ($k_P(T_2)/k_P(T_1)$ on the order of 10$^2$-10$^3$, *vide infra*), which makes the direct observation of RTP(T$_2$) possible.

To further study the unusual photophysical behaviors of the BPBF$_2$-PhB systems, two more BPBF$_2$ compounds, **2** and **3**, are synthesized; their structural characterization results and photophysical data are attached in Supplementary Information. Upon doping into PhB matrices, **2**-PhB-0.1% samples show fluorescence band in the range of 400 to 600 nm with $\lambda_F$ of 432 nm (2.87 eV) in their steady-state emission spectra (Fig. 2a and Supplementary Fig. 10). The delayed emission spectra of **2**-PhB-0.1% samples at room temperature exhibit two clearly resolved phosphorescence bands with maxima at 421 nm (2.95 eV) and 474 nm (2.62 eV), respectively; again, the wavelength of the emission maxima of the higher-energy bands (421 nm) is shorter than that of the fluorescence bands ($\lambda_F$ = 432 nm). TD-DFT calculation reveals that the T$_2$ state of compound **2** at optimized T$_2$ geometry possesses typical $^3$n-π* character from benzophenone moieties (Fig. 2c). The T$_2$ to S$_0$ phosphorescence decay at optimized T$_2$ geometry has a large SOCME of 44.09 cm$^{-1}$ (Fig. 2c and Supplementary Fig. 11) ($k_P(T_2)/k_P(T_1)$ on the order of 10$^3$, *vide infra*), which supports the

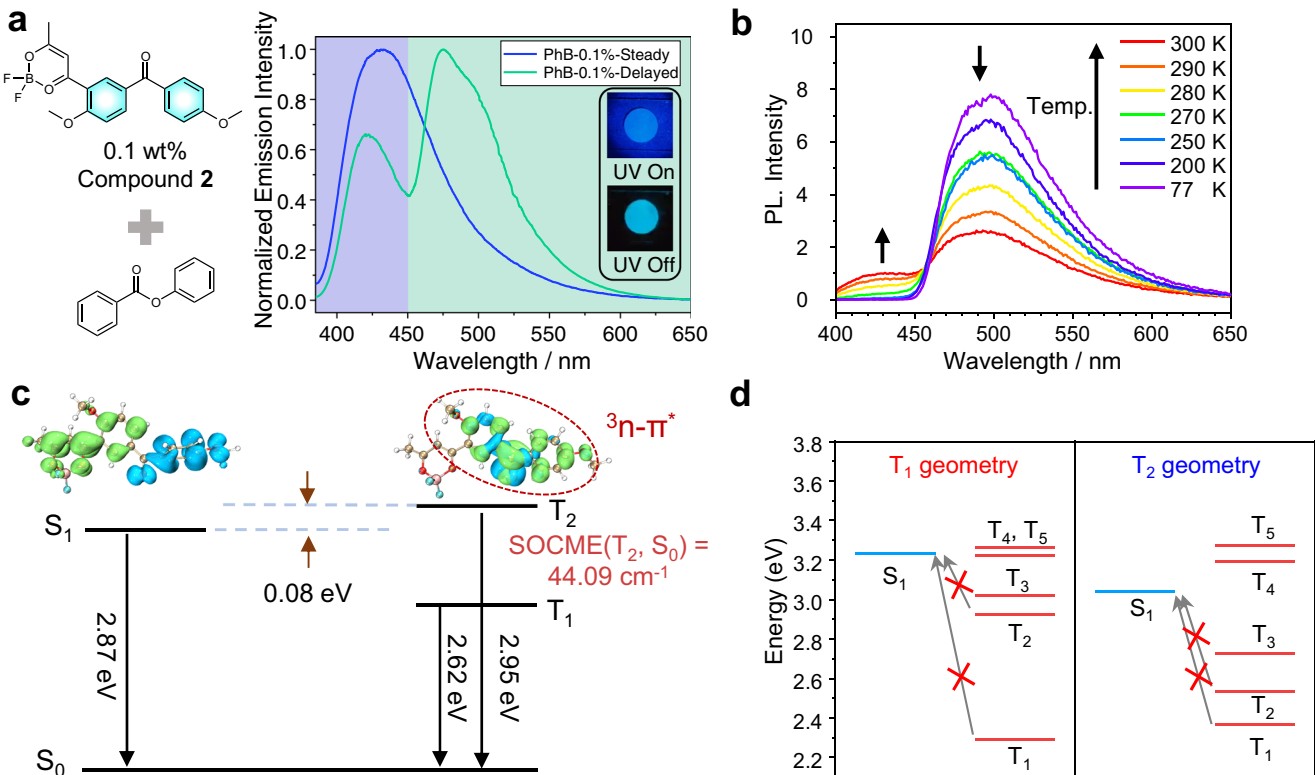

**Fig. 2 | Photophysical properties of compound 2 system. a** Molecular structure of compound **2**, steady-state and delayed emission spectra (1 ms delay) of **2**-PhB-0.1% powder at room temperature, and fluorescence and afterglow photographs of **2**-PhB-0.1% powder; **b** Temperature-dependent delayed emission spectra (1 ms delay) of **2**-PhB-0.1% powder from 77 K to 300 K; **c** Experimental and TD-DFT calculation results of $S_1$, $T_1$, $T_2$ states of compound **2**: the isosurface maps of electron-hole density difference, SOCME values (at optimized $T_2$ geometry) and energy levels estimated from emission maxima. TD-DFT calculations were performed on ORCA 5.0.3 program with B3LYP/G functional and def2-TZVP(-f) basis set; **d** Excited state energy levels at the optimized geometry of $T_1$ and $T_2$ states of compound **2**, respectively.

existence of dual RTP behaviors in **2**-PhB systems. Unlike the reported dual RTP systems[26–32], the energy levels of the emissive $T_2$ states (2.95 eV, estimated from the emission maxima) are higher than those of the $S_1$ states (2.87 eV, estimated from the fluorescence maxima). Here experimental $\Delta E(S_1$-$T_2)$ (−0.08 eV) and $\Delta E(S_1$-$T_1)$ (0.25 eV) for forward ISC are relatively small. The TD-DFT calculated $\Delta E(S_1$-$T_2)$ (−0.23 eV) and $\Delta E(S_1$-$T_1)$ (0.02 eV) for forward ISC at the optimized geometry of $S_1$ state are also relatively small (Supplementary Table 3). Besides, the $S_1$ states have different symmetry from both $T_1$ and $T_2$ states (Supplementary Fig. 11). According to the energy gap law and the El-Sayed rule, the **2**-PhB system should possess strong tendency to undergo forward ISC. It is noteworthy that, for the reverse ISC where the excited state energy levels are calculated at the optimized geometry of $T_1$ and $T_2$ states, both $T_1$ and $T_2$ levels of compound **2** are calculated by TD-DFT to be much lower than the $S_1$ state (Fig. 2d). Given that reverse ISC starts from triplet excited states, these suggest that the reverse ISC and subsequent TADF would be unlikely to occur in **2**-PhB system; these theoretical analyses agree with the experimental results where the room-temperature delayed emission spectra of **2**-PhB materials show the absence of TADF signals that can coincide with the 432 nm fluorescence band (Fig. 2a). Variable temperature phosphorescence measurements (1 ms delay) have been performed. Figure 2b displays that $T_1$ phosphorescence band in the lower-energy region dominates at 77 K; $T_2$ phosphorescence signal in the higher-energy region has not been observed at 77 K. Upon increasing temperature, the $T_2$ phosphorescence bands in the range of 400 to 450 nm appear at 270 K and are found to increase with temperature (Fig. 2b). Figure 1f also exhibits the emergence of $T_2$ phosphorescence upon increasing temperature in **1**-PhB system. In both of **1**-PhB (Fig. 1f) and

**2**-PhB systems, the $S_1$ to $T_2$ ISC and $T_1$ to $T_2$ reverse internal conversion are up-conversion processes whose speed increase with temperature. Therefore, the variable temperature delayed emission studies suggest that the $T_2$ states of $^3n$-$\pi^*$ characters in both **1**-PhB and **2**-PhB systems are populated by thermally activated ISC process from $S_1$ states of $^1CT$ characters and $T_1$ to $T_2$ reverse internal conversion.

In the case of **3**-PhB systems, the up-converted RTP bands at 424 nm (2.92 eV) become the main emission signals in the delayed emission spectra (1 ms delay); the steady-state emission spectra show fluorescence bands at 434 nm (2.86 eV) (Fig. 3a and Supplementary Fig. 12). The **3**-PhB samples at ambient conditions show blue emission under 365 nm UV lamp, and exhibit deep-blue afterglow upon ceasing the 365 nm UV lamp (Fig. 3a). At 77 K, the higher-energy phosphorescence bands at 424 nm (2.92 eV) disappear in the delayed emission spectra, while the lower-energy phosphorescence bands observed at 465 nm (2.67 eV) are assigned as radiative decay of $T_1$ states. Variable temperature phosphorescence measurements (1 ms delay) show the enhancement of higher-energy phosphorescence bands at 424 nm upon temperature increase (Fig. 3b). TD-DFT calculations of excited state energy levels at the optimized geometry of $T_1$, $T_2$, and $T_3$ states show that **3**'s $S_1$ level is much higher than $T_1$, $T_2$ and $T_3$ states (Fig. 3c), which suggest reverse ISC and TADF should be insignificant in **3**-PhB system. From TD-DFT calculation, it has been found that the $T_3$ state of compound **3** has significant n-$\pi^*$ transition character from benzophenone group (Fig. 3c), exhibiting $T_3$-$S_0$ SOCME of 24.44 cm$^{-1}$ (Supplementary Fig. 13). Besides, $k_P(T_3)$ has been obtained by theoretical calculation to be on the order of $10^3$-$10^4$ s$^{-1}$, much larger than $k_P(T_2)$ and $k_P(T_1)$ (*vide infra*). These suggest that the up-converted deep-blue RTP band at 424 nm originates from $T_3$ to $S_0$ phosphorescence decay;

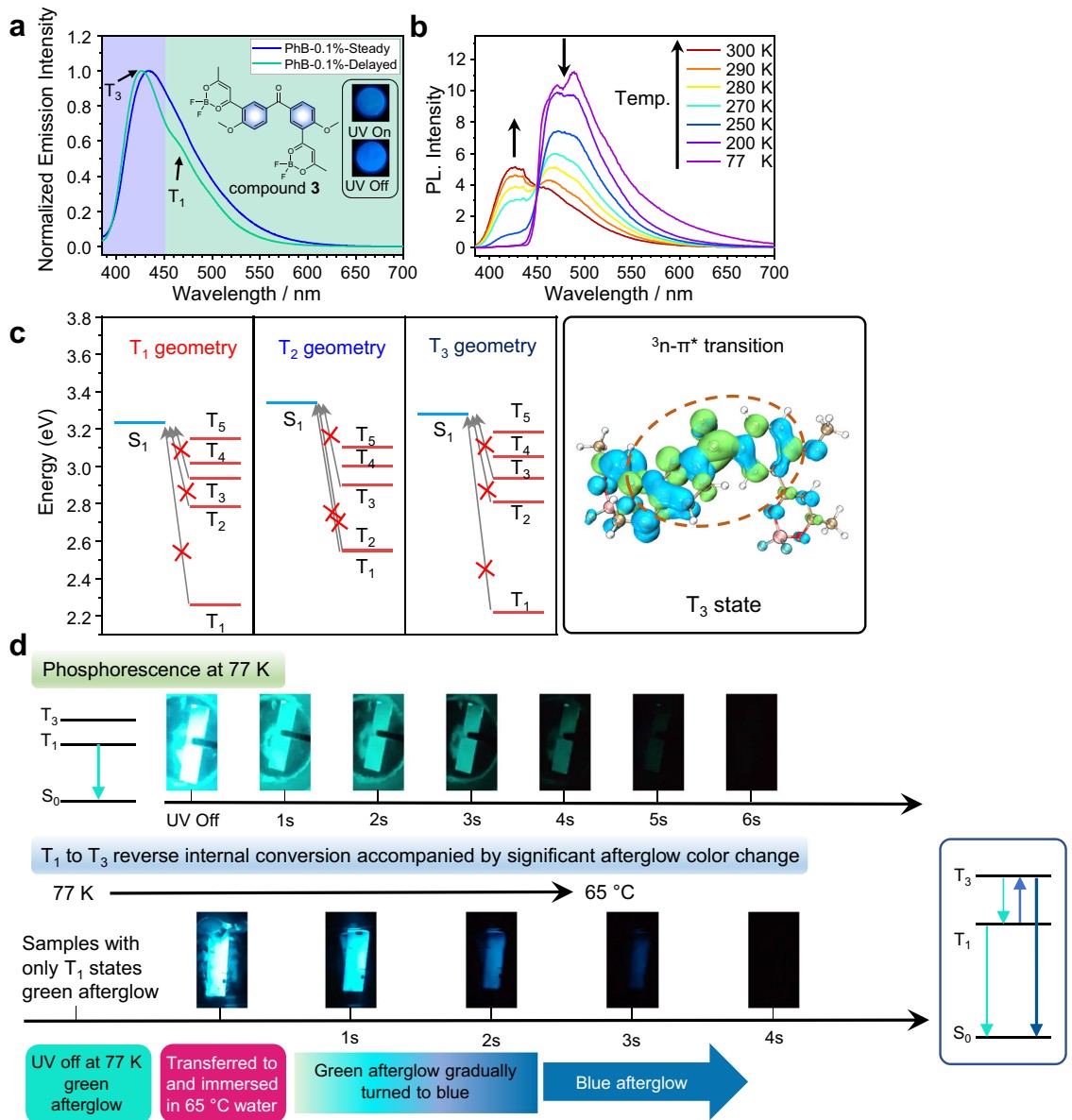

**Fig. 3 | Photophysical properties of compound 3 system. a** Molecular structure of compound **3**, steady-state and delayed emission spectra (1 ms delay) of **3**-PhB-0.1% powder at room temperature, and fluorescence and afterglow photographs of **3**-PhB-0.1% powder at room temperature; **b** Temperature-dependent delayed emission spectra (1 ms delay) of **3**-PhB-0.1% powder from 77 K to 300 K; **c** Excited state energy levels at the optimized geometry of $T_1$, $T_2$ and $T_3$ states of compound **3**, respectively. The right picture is the isosurface maps of electron-hole density difference of **3**'s $T_3$ state calculated at the $S_0$ geometry with B3LYP/G functional and def2-TZVP(-f) basis set; **d** Phosphorescence photographs of melt-cast film of **3**-PhB-0.1% at 77 K, photographs showing the phenomenon of afterglow color change when transferring **3**-PhB-0.1% melt-cast film from liquid nitrogen (77 K) to hot water (65 °C) and the proposed mechanism.

**Table 1 | Photophysical data of BPBF₂-PhB materials under ambient conditions**

| Entry | $\lambda_F(S_1)$ /nm | $\lambda_P(T_n)$ [a] /nm | $\tau_{avg}(T_n)$ [a] /ms | $\lambda_P(T_1)$ /nm | $\tau_{avg}(T_1)$ /ms | PLQY |
|---|---|---|---|---|---|---|
| **1**-PhB-0.1% powder | 437 | 421 | 302.6 | 483 | 328.6 | 16.5% |
| **2**-PhB-0.1% powder | 432 | 421 | 278.2 | 474 | 297.9 | 9.4% |
| **3**-PhB-0.1% powder | 434 | 424 | 180.1 | 465 | 286.2 | 12.4% |

[a] $n = 2$ for **1**-PhB and **2**-PhB materials, $n = 3$ for **3**-PhB material.

the $T_3$ states should be populated from $S_1$ to $T_3$ ISC and $T_1$ to $T_3$ reverse internal conversion.

Table 1 summarizes the photophysical data of BPBF₂-PhB materials under ambient conditions. It is noteworthy that the phosphorescence lifetimes of both $T_1$-$S_0$ and $T_n$-$S_0$ are longer than 0.1 s (Table 1 and Supplementary Fig. 14). From $\tau_{avg}(T_1)$ of several hundred

milliseconds, $k_P(T_1)$ can be estimated to be on the order of $10^0$-$10^1$ s⁻¹. Unlike the slow $T_1$-$S_0$ phosphorescence decay, $T_n$ to $S_0$ phosphorescence process should have much large $k_P(T_n)$ of around $10^3$ s⁻¹ ($T_n$ states have significant n-π* transition characters). Therefore, the photophysical pathway of $S_1$ to $T_n$ ISC and subsequent $T_n$ to $S_0$ phosphorescence decay cannot directly explain the long $\tau_{avg}(T_n)$ of several

**Table 2 | $S_1$, $T_1$ and $T_n$ excitation energies (eV) of BPBF$_2$ compounds calculated at respective optimized geometry ($n$ = 2 for compound 1 and 2, $n$ = 3 for compound 3)**

| Method | State | $S_1$ geometry | $T_1$ geometry | $T_n$ geometry |
|---|---|---|---|---|
| TD-B3LYP/G/def2-TZVP(-f) | **Compound 1** | | | |
| | $S_1$ | 2.368 | 3.379 | 3.264 |
| | $T_1$ | 2.297 | 2.266 | 2.554 |
| | $T_n$ | 2.711 | 2.938 | 2.683 |
| | **Compound 2** | | | |
| | $S_1$ | 1.981 | 3.231 | 3.042 |
| | $T_1$ | 1.963 | 2.293 | 2.367 |
| | $T_n$ | 2.213 | 2.924 | 2.536 |
| | **Compound 3** | | | |
| | $S_1$ | 2.192 | 3.233 | 3.281 |
| | $T_1$ | 2.155 | 2.258 | 2.218 |
| | $T_n$ | 2.517 | 2.934 | 2.936 |
| TD-ωB97X-D3/def2-TZVP(-f) | **Compound 1** | | | |
| | $S_1$ | 3.135 | 3.578 | 3.480 |
| | $T_1$ | 2.472 | 2.307 | 2.732 |
| | $T_n$ | 2.898 | 3.121 | 2.847 |
| | **Compound 2** | | | |
| | $S_1$ | 3.180 | 3.856 | 3.111 |
| | $T_1$ | 2.612 | 2.448 | 2.494 |
| | $T_n$ | 2.966 | 3.273 | 2.969 |
| | **Compound 3** | | | |
| | $S_1$ | 3.167 | 3.762 | 3.591 |
| | $T_1$ | 2.600 | 2.392 | 2.897 |
| | $T_n$ | 3.071 | 3.258 | 3.002 |

**Table 3 | Calculated SOCME (cm$^{-1}$), ISC rate ($k_{ISC}$, s$^{-1}$), and fluorescence emission rate ($k_F$, s$^{-1}$) at the optimized $S_1$ geometry of 1–3 with ωB97X-D3 functional and def2-TZVP(-f) basis set**

| | SOCME (cm$^{-1}$) | | $k_{ISC}$ (s$^{-1}$) | | $k_F$ (s$^{-1}$) |
|---|---|---|---|---|---|
| | $S_1 \rightarrow T_1$ | $S_1 \rightarrow T_n{}^a$ | $S_1 \rightarrow T_1$ | $S_1 \rightarrow T_n{}^a$ | $S_1 \rightarrow S_0$ |
| 1 | 21.92 | 26.18 | 3.70*10$^9$ (1.86*10$^9$) $^b$ | 1.50*10$^{10}$ (2.70*10$^7$) $^b$ | 2.19*10$^7$ |
| 2 | 11.44 | 6.68 | 1.55*10$^9$ (3.92*10$^{10}$) $^b$ | 2.51*10$^9$ (5.89*10$^9$) $^b$ | 5.95*10$^7$ |
| 3 | 12.85 | 9.59 | 1.83*10$^9$ (5.04*10$^{10}$) $^b$ | 2.63*10$^9$ (4.84*10$^8$) $^b$ | 7.47*10$^7$ |

$^a$$n$ = 2 for compound **1** and **2**, $n$ = 3 for compound **3**.
$^b$Intersystem crossing rate at 300 K calculated by Marcus theory.

hundred milliseconds. We propose the existence of thermally activated $T_1$ to $T_n$ reverse internal conversion in BPBF$_2$-PhB systems under ambient conditions[26,31,32]. The $T_1$-$T_n$ equilibrium under ambient conditions can explain the observed long $\tau_{avg}(T_n)$ of several hundred milliseconds, since the long-lived $T_1$ states can serve as reservoir for RTP($T_n$). To verify this, we first prepare a **3**-PhB sample which emit green afterglow at 77 K upon ceasing UV excitation source; in this sample, only $T_1$ states exist upon switching off UV excitation (Fig. 3d). After being immediately transferred to and immersed in a 65 °C water bath, the **3**-PhB sample show afterglow color change from green to blue (Fig. 3d). The blue afterglow emission of this sample can be exclusively attributed to $T_n$-$S_0$ phosphorescence, rather than TADF; in **3**-PhB systems, TADF is insignificant as discussed above (Fig. 3a, b). These observations in Fig. 3d provide very strong evidence on the presence of thermally activated $T_1$ to $T_n$ reverse internal conversion in the present systems. Such $T_1$ to $T_n$ reverse internal conversion accompanied by significant afterglow color change visible by naked

eyes, which have not been reported in the literature[26,31,32], can provide straightforward understanding on the triplet excited state dynamics in organic systems. From Table 1, $T_n$-$T_1$ energy gap can be estimated from phosphorescence maxima to be 0.38 eV, 0.33 eV, and 0.26 eV for **1**-PhB-0.1% ($n$ = 2), **2**-PhB-0.1% ($n$ = 2) and **3**-PhB-0.1% ($n$ = 3) powders, respectively. The decrease of $T_n$-$T_1$ energy gap can give rise to the increase of the population of $T_n$ states, which is in line with the increase of RTP($T_n$)/RTP($T_1$) intensity ratios in these BPBF$_2$-PhB-0.1% powder samples under ambient conditions (Fig. 1c, 2a and 3a).

**Theoretical investigations**

Theoretical calculations on the energy level structures, intersystem crossing, internal conversion, and radiative decay have been performed to further study the intriguing photophysical behaviors in the present system. Table 2 summarizes the excited state energy levels calculated by TD-B3LYP/def2-tzvp(-f) method for the $S_0$, $S_1$, $T_1$, and $T_n$ geometries of compounds **1** to **3**; optimization of these geometry was performed at TD-B3LYP/G/6-31G(d,p) level of theory (Supplementary Tables 2–4, Supplementary Tables 12–14). It has been found that, at the optimized geometry of $S_1$, the energy levels of $S_1$ states are sandwiched between $T_1$ and $T_n$ states ($n$ = 2 for compounds **1** and **2**, $n$ = 3 for compound **3**), which agree with experimental observations (Fig. 1c, Fig. 2a, Fig. 3a and Table 1). It is known that, in TD-DFT calculation, the use of hybrid functionals such as B3LYP can reduce self-interaction error but does not eliminate it[44,45]. Range-separated hybrid functionals have been reported to mitigate the systematic error[46,47], and recent studies showed that the range-separated ωB97X-D functional[48] exhibits better overall performance on modeling electronically excited states compared to B3LYP functional[49,50]. Here advanced method of TD-ωB97X-D3/def2-tzvp(-f) that may rule out the systematic error coming from B3LYP have also been used to calculate excited state energy levels, which has also been summarized in Table 2, Supplementary Tables 8–10, and Supplementary Tables 15–17; geometry optimization for compound **1**–**3** was performed at ωB97XD/6-31G(d,p) level of theory. It is found that the energy levels obtained by TD-ωB97X-D3/def2-TZVP(-f) calculation are relatively close to those by experimental observations (Supplementary Tables 2–10 and 11), so we use the results obtained by TD-ωB97X-D3/def2-TZVP(-f) method for the quantum mechanical Fermi's golden rule (FGR) rate calculation in the present study.

For the forward ISC, SOCME values of $S_1$ to $T_n$ for different geometries have been calculated on ORCA 5.0.3 program with spin-orbit mean-field (SOMF) methods at ωB97X-D3/def2-tzvp(-f) level. Table 3 and Supplementary Tables 15–17 show that most ISC channels have SOCME above 1 cm$^{-1}$ and some ISC channels possess SOCME larger than 10 cm$^{-1}$, which suggest the strong tendency of forward ISC in the system. In the literature, Kaji and coworkers reported the theoretical calculation of quantitative rates of the photophysical processes in benzophenone systems[51,52]. Here the luminescent compounds contain benzophenone functional groups, so we use Kaji's method to calculate ISC rate constants based on the FGR rate theory (computational details in Supplementary Methods). The calculated ISC rate constants have been summarized in Table 3 and Supplementary Tables 24–26. Besides, the calculations of ISC rate constants via Marcus theory have also been performed[53,54] (Table 3 and Supplementary Tables 27–29). Both FGR and Marcus theory show that the ISC rate constants of $S_1$-to-$T_1$ and $S_1$-to-$T_n$ ($n$ = 2 for compounds **1** and **2**, $n$ = 3 for compound **3**) are above $10^7$ s$^{-1}$ (Table 3). Given the fluorescence decay of $S_1$ states of intramolecular charge transfer nature has rate constants of $10^7$ to $10^8$ s$^{-1}$ (Table 3), such large ISC rate constants would result in relatively high ISC quantum yields in the system. Actually, in the experimental studies, the steady-state emission spectra of **1**-**3** solutions in dichloromethane at 77 K exhibit significant components of phosphorescence signals (Supplementary Fig. 15), which also support the strong tendency of ISC in the present system. For the reverse ISC, from the results

obtained by both B3LYP and ωB97X-D3 methods, it is found that, at the optimized geometry of either $T_1$ or $T_n$ ($n = 2$ for compounds **1** and **2**, $n = 3$ for compound **3**), the $T_1$ and $T_n$ levels are much lower than $S_1$ levels (Table 2). Given that reverse ISC starts from triplet excited states, these results suggest that reverse ISC is not likely to occur. The corresponding rate constants of reverse ISC have also been calculated to show small values (Supplementary Tables 24–29), which can explain the absence of TADF afterglow in the experimental observations.

**Table 4 | Internal conversion rates ($k_{IC}$, s$^{-1}$) between $T_1$ and $T_n$ of 1–3 calculated by different methods ($n = 2$ for compound 1 and 2, $n = 3$ for compound 3). Geometries and frequencies are calculated at TD-ωB97XD/6-31G(d,p) level of theory, and energy differences are calculated at TD-ωB97X-D3/def2-TZVP(-f) level of theory**

|  | $S_O$ geometry | $S_1$ geometry | $T_1$ geometry | $T_n$ geometry | FCclasses[a] |
|---|---|---|---|---|---|
| **Compound 1** | | | | | |
| $k_{IC}(T_n{}^a{\rightarrow}T_1)$ | 1.43*10$^{12}$ | 3.93*10$^{11}$ | 1.08*10$^{11}$ | 5.39*10$^{12}$ | 3.02*10$^{10}$ |
| $k_{IC}(T_1{\rightarrow}T_n{}^a)$ | 2.57*10$^{8}$ | 1.00*10$^{5}$ | 2.28*10$^{-3}$ | 6.31*10$^{10}$ | 2.02*10$^{8}$ |
| **Compound 2** | | | | | |
| $k_{IC}(T_n{}^a{\rightarrow}T_1)$ | 2.20*10$^{10}$ | 1.83*10$^{10}$ | 3.37*10$^{9}$ | 1.02*10$^{10}$ | 2.67*10$^{11}$ |
| $k_{IC}(T_1{\rightarrow}T_n{}^a)$ | 8.25*10$^{4}$ | 2.49*10$^{4}$ | 4.66*10$^{-5}$ | 1.07*10$^{2}$ | 7.30*10$^{8}$ |
| **Compound 3** | | | | | |
| $k_{IC}(T_n{}^a{\rightarrow}T_1)$ | 2.61*10$^{12}$ | 1.18*10$^{12}$ | 3.48*10$^{11}$ | 2.37*10$^{13}$ | 6.50*10$^{10}$ |
| $k_{IC}(T_1{\rightarrow}T_n{}^a)$ | 1.28*10$^{7}$ | 1.44*10$^{4}$ | 9.85*10$^{-4}$ | 4.08*10$^{11}$ | 1.20*10$^{8}$ |

[a]The rate calculations were performed with FCclasses3[58]. Detailed information can be found in Supplementary Information.
Temperature was set to 300 K

**Table 5 | Calculated SOCME (cm$^{-1}$), transition dipole moments (TDM, a.u.), and phosphorescence emission rate ($k_p$, s$^{-1}$) of 1–3 based on optimized triplet excited states geometries at the ωB97X-D3/def2-TZVP(-f) level of theory**

|  | SOCME (cm$^{-1}$) | | TDM (a. u.) | | $k_P$ (s$^{-1}$) | |
|---|---|---|---|---|---|---|
|  | $T_1 \rightarrow S_O$ | $T_n{}^a \rightarrow S_O$ | $T_1 \rightarrow S_O$ | $T_n{}^a \rightarrow S_O$ | $T_1 \rightarrow S_O$ | $T_n{}^a \rightarrow S_O$ |
| 1 | 0.94 | 32.61 | 1.8439*10$^{-4}$ | 2.8084*10$^{-3}$ | 2.86 | 9.69 *10$^{2}$ |
| 2 | 1.20 | 26.09 | 3.1097*10$^{-4}$ | 7.6159*10$^{-3}$ | 1.06 *10$^{1}$ | 9.28 *10$^{3}$ |
| 3 | 0.52 | 36.05 | 4.1461*10$^{-4}$ | 3.1489*10$^{-3}$ | 1.76 *10$^{1}$ | 1.35 *10$^{3}$ |

[a]$n = 2$ for compound **1** and **2**, and $n = 3$ for compound **3**.

Upon forward ISC, we propose that, based on the experimental observations summarized in Table 1 and shown in Fig. 3d, the resultant $T_1$ and $T_n$ states would build $T_1$-$T_n$ equilibrium via forward and reverse internal conversion. Kaji's method[51,52] has been used to calculate the rate constants of forward internal conversion; frequency analyses are performed at ωB97XD/6-31G(d,p) level of theory (detailed in Supplementary Methods). It is found that forward internal conversion is very fast with rate constant of $k(T_n$-$T_1)$ on the order of 10$^9$-10$^{13}$ s$^{-1}$ (Table 4). For the reverse internal conversion, the rate constants $k(T_1$-$T_n)$ calculated by the Arrhenius-type expression (See Supplementary Equation (14)) are found to be largely underestimated (Table 4). Recent studies of anti-Kasha systems[55–57] showed that electron-vibrational coupling should be taken into account for the calculation of both $k(T_n$-$T_1)$ and $k(T_1$-$T_n)$. Accordingly, based on the FCclasses software[58], $k(T_n$-$T_1)$ and $k(T_1$-$T_n)$ have been calculated to be 10$^{10}$-10$^{11}$ s$^{-1}$ and 10$^8$-10$^9$ s$^{-1}$, respectively (Table 4 and Supplementary Table 30). The $k(T_1$-$T_n)/k(T_n$-$T_1)$ ratios have been calculated to be on the order of 10$^{-3}$-10$^{-2}$ at 300 K (Table 4), much larger than those estimated by the Arrhenius-type expression (Table 4). To investigate the phosphorescence decay of $T_1$ and $T_n$ states, the corresponding SOCME values, transition dipole moments, and phosphorescence rate constants have been calculated and summarized in Table 5. Calculated at TD-ωB97X-D3/def2-tzvp(-f) level of theory, the SOCME values and transition dipole moments of $T_n$-$S_O$ phosphorescence decay at $T_n$ geometries ($n = 2$ for compounds **1** and **2**, $n = 3$ for compound **3**) have been found to be much larger than those of $T_1$-$S_O$ phosphorescence decay at $T_1$ geometries (Table 5). Phosphorescence rate constants, $k_P(T_1)$ and $k_P(T_n)$, have been obtained by FGR rate theory (Table 5). The $k_P(T_n)$ values of 10$^3$-10$^4$ s$^{-1}$ at $T_n$ geometries ($n = 2$ for compounds **1** and **2**, $n = 3$ for compound **3**) are found to be much larger than $k_P(T_1)$ values of 10$^0$-10$^1$ s$^{-1}$ at $T_1$ geometries, exhibiting $k_P(T_n)/k_P(T_1)$ ratios of 10$^2$-10$^3$. Given that the relative emission intensity of RTP($T_n$)/RTP($T_1$) is proportional to $k(T_1$-$T_n)/k(T_n$-$T_1) \times k_P(T_n)/k_P(T_1)$, the above theoretical calculations support the experimental observation of RTP($T_n$)/RTP($T_1$) dual emission in the delayed spectra.

Figure 4a illustrates the photophysical mechanism of the dual RTP systems. Upon excitation, $S_1$ states of intramolecular charge transfer character form. Because of the involvement of benzophenone functional groups, different symmetry between $S_1$ states and triplet excited states, and relatively small singlet-triplet splitting energies, the system shows strong tendency to undergo intersystem crossing to form $T_1$ and $T_n$ states. Upon intersystem crossing, the $T_1$ and $T_n$ states build equilibrium under ambient conditions due to the fast internal conversion and reverse internal conversion facilitated by electron-vibrational coupling. The $T_n$ states of n-π* characters from benzophenone groups ($n = 2$ for compounds **1** and **2**, $n = 3$ for compound **3**) have large

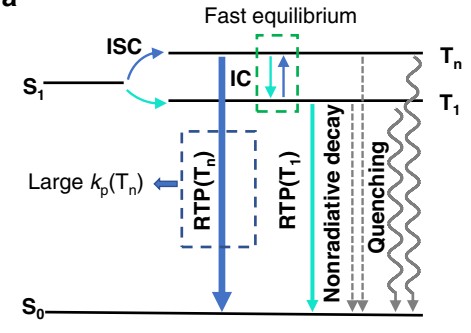

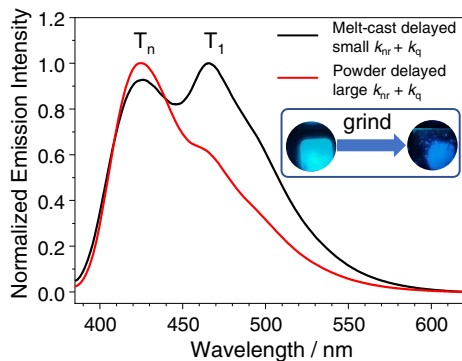

**Fig. 4 | The proposed dual RTP mechanism and RTP($T_n$)/RTP($T_1$) ratiometric response towards mechanical grinding. a** Jablonski diagram of BPBF$_2$-PhB afterglow system; **b** Delayed emission spectra (1 ms delay) of **3**-PhB-0.1% melt-cast sample (black line, with small $k_{nr} + k_q$ value) and powder sample (red line, with large $k_{nr} + k_q$ value) and the phosphorescence photographs after ceasing excitation light source (UV off, 0.1 s) under ambient conditions.

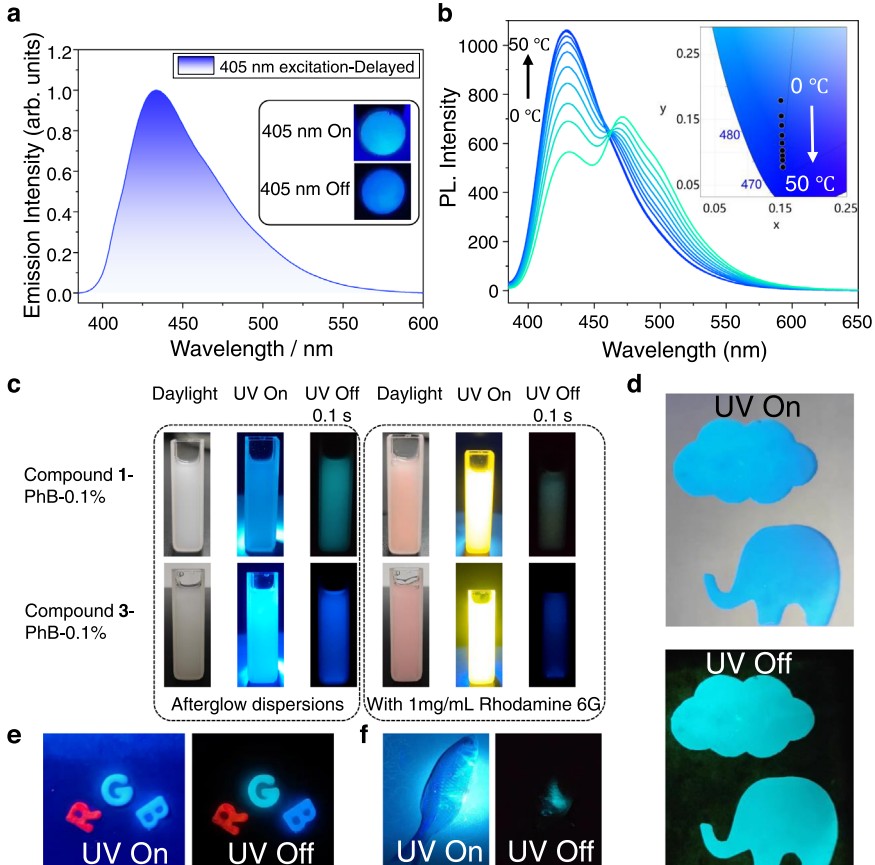

**Fig. 5 | Functionalities of the up-converted RTP materials. a** Delayed emission spectra (1 ms delay) of **3**-PhB-0.1% powder excited by 405 nm at room temperature. Inset: photographs of the **3**-PhB-0.1% powder, recorded upon switching on (top) and off (bottom) a 405 nm laser; **b** Temperature-dependent delayed emission spectra (1 ms delay) of **3**-PhB-0.1% powder from 0 to 50 °C and CIE diagram; **c** Aqueous afterglow dispersion of BPBF$_2$-PhB-0.1% stabilized by Pluronic F127 surfactant; **d** Afterglow pattern of **1**-PhB-0.1% obtained by UV excitation through pre-designed mask; **e** RGB-colored afterglow objects, the letters R, G, and B, were made from NPhRedBF$_2$−MeOBP-0.1%[62], **1**-PhB-0.1% powder, and **3**-PhB-0.1% powder, respectively; **f** Preliminary bioimaging experiments of the aqueous dispersion of **1**-PhB material in fish.

phosphorescence decay rates and counterbalance the small population of $T_n$ states, leading to RTP($T_n$) emission that violate Kasha's rule. The PhB matrices can provide rigid microenvironment to suppress nonradiative decay ($k_{nr}$) and oxygen quenching ($k_q$) of BPBF$_2$'s triplet excited states. The $T_1$ states have small phosphorescence rates of $10^0$-$10^1$ s$^{-1}$ to show afterglow property with lifetimes of several hundred milliseconds under ambient conditions. Because of the long-lived excited state nature, the $T_1$ states also serve as store for the population of $T_n$ states via $T_1$-$T_n$ equilibrium, which is the reason that RTP($T_n$) also possess lifetimes of several hundred milliseconds.

**Material functions**

In view of the large difference between $k_P(T_1)$ and $k_P(T_n)$, we conceive that the dual RTP materials would give RTP($T_n$)/RTP($T_1$) ratiometric response towards the change of microenvironment of triplet excited states. In the system of **3**-PhB-0.1% under ambient conditions, it is found that powder samples after mechanical grinding have large RTP($T_3$)/RTP($T_1$) intensity ratio than melt-cast samples in the room-temperature phosphorescence spectra (Fig. 4b). Powder samples have large $k_{nr} + k_q$ values than melt-cast samples given other conditions being fixed, because powders have larger surface area exposed to air and sometimes low crystallinity than melt-cast samples. In the reported studies of single-component luminescent systems[59–61], upon grinding crystalline samples into powders, the emission spectra showed significant change because of the change of aggregation structures in the single-component systems. This is not the case in the present study, since in **3**-PhB-0.1% samples at such a low doping concentration, most of **3** molecules are in monomeric form rather than in aggregation state. The mechano-responsive RTP property of **3**-PhB-0.1% materials derives from the very different $k_P$ values between $T_1$ states and $T_3$ states and consequently significant RTP($T_3$)/RTP($T_1$) ratiometric enhancement is observed upon increasing $k_{nr} + k_q$ values.

Because of the room-temperature phosphorescence from higher triplet excited states, the present study of **3**-PhB system with $\lambda_P(T_3) < \lambda_F(S_1)$ and thus small Stokes shift provides a unique method to achieve deep-blue afterglow at room temperature by using UVA or visible light excitation; the deep-blue afterglow under ambient conditions can also be obtained by exciting the **3**-PhB samples at 405 nm (Fig. 5a). In contrast, the reported studies based on conventional RTP mechanism (with large Stokes shift) for deep-blue afterglow requires the use of UV excitation at 310 nm or 280 nm or even shorter wavelengths[5,9]. Since both $S_1$ to $T_3$ ISC and $T_1$ to $T_3$ reverse internal conversion are uphill processes and temperature dependent, the **3**-PhB afterglow materials can function as temperature sensor in the range of 0 to 50 °C as shown in Fig. 5b and Supplementary Fig. 16. This temperature-responsive property is originated from the different population mechanisms between $T_1$ states and $T_3$ states. The BPBF$_2$-PhB afterglow materials have been found to be readily melt-cast into large-area films and various shaped objects and processed into aqueous dispersion with the aid of Pluronic F127 surfactants (Fig. 5c). Diverse patterns can be obtained by UV excitation through pre-designed masks (Fig. 5d). Combined with other afterglow materials, RGB-colored afterglow objects can be obtained which can be used to increase security levels of anti-counterfeiting techniques (Fig. 5e).

Figure 5c shows the aqueous dispersion of afterglow materials exhibit significant blue emission after ceasing UV lamp, which can eliminate the interference of strong background fluorescence. Preliminary in vivo bioimaging studies has also been performed to display very clean background in the afterglow imaging mode after switching off the excitation source (Fig. 5f).

## Discussion

In summary, the present study reports a serendipitous finding of up-converted RTP with $\lambda_P(T_n) < \lambda_F(S_1)$ and $\tau_P > 0.1$ s in $BPBF_2$-PhB systems, which has been rarely observed in the reported studies. The involvement of benzophenone functional groups on $BPBF_2$ molecules is very important to achieve such up-converted RTP in the dopant-matrix systems since it not only facilitates ISC but also endows $T_n$ ($n \geq 2$) states with n-π* character and large $k_P$. Given that the energy levels of the $T_n$ states are mainly determined by the benzophenone groups, here the use of difluoroboron β-diketonate functional groups (with suitable LUMO level and electron-accepting strength) is also very important to result in a proper $\Delta E(T_n$-$T_1)$ in $BPBF_2$ system.

The present study shows that it is still possible to form $T_1$-$T_n$ equilibrium under ambient conditions in organic systems with $\Delta E(T_n$-$T_1)$ of around 0.3 eV. Theoretical studies reveal that the electron-vibrational coupling can increase the population of $T_n$ states, and the large $k_P(T_n)/k_P(T_1)$ ratios can compensate the small population of $T_n$ states, leading to anti-Kasha RTP($T_n$) emission. Here the clearly resolved RTP($T_n$) and RTP($T_1$) bands endow the $BPBF_2$-PhB materials with stimuli-responsive functions via RTP($T_n$)/RTP($T_1$) ratiometric change towards mechanical force and temperature variation.

The change of RTP($T_n$) emission reflects the change of photophysical processes related to $T_n$ states, so the direct observation of RTP($T_n$) facilitates the study of the population, equilibrium, and radiative decay of $T_n$ states. The present study would have significant impact on the deep understanding of photophysical behaviors of higher triplet excited states and provide strategies for designing high-performance organic afterglow materials with intriguing properties.

## Methods

### Physical measurements and instrumentation

Nuclear magnetic resonance (NMR) spectra were recorded on a JEOL Fourier-transform NMR spectrometer (400 MHz), including [1]H NMR, [13]C NMR, [19]F NMR, [11]B NMR. Mass spectra were performed on Agilent Technologies 5973 N and Thermo Fisher Scientific LTQ FT Ultras mass spectrometer. FT-IR spectra were recorded on a Nicolet AVATAR-360 FT-IR spectrophotomerter with a resolution of 4 cm$^{-1}$. Single-crystal X-ray diffraction analysis was performed on a D8 VENTURE SC-XRD instrument. UV-Vis absorption spectra were recorded on a Techcomp UV1050 UV-vis spectrophotometer. Emission spectra were recorded using Hitachi FL-4700 fluorescence spectrometer, Hitachi FL-7000 fluorescence spectrometer and Horiba FluoroLog-3 fluorescence spectrometer. Photoluminescence quantum yield was measured by a Hamamatsu absolute PL quantum yield measurement system based on a standard protocol. Photographs and videos were captured by Xiaomi 11 Ultra camera. Before the capture, samples were irradiated by a 365 nm UV lamp (5 W) for approximately 5 s at a distance of approximately 15 cm.

### Synthesis of luminescent compounds via cascade reaction

In a round bottom flask, boron trifluoride diethyl etherate (2.0 mL, 15.8 mmol) was slowly added into a stirred solution of 4-methoxybenzophenone (425 mg, 2.00 mmol) in acetic anhydride (5.00 mL, 51.4 mmol). The reaction mixture was kept at 80 °C and stirred for 12 h. Then the reaction was quenched by dropwisely adding the reaction mixture into cold water. The precipitates were washed by deionized water for three times and dried under vacuum. The crude product of compound **1** was obtained by column chromatography over silica gel using petroleum ether/dichloromethane (1:2) as eluent. The product of compound **1** was further purified by three times recrystallization in spectroscopic grade dichloromethane/hexane, giving a pale yellow solids with an isolation yield of 22.6% (156 mg). [1]H NMR (400 MHz, Chloroform-*d*) δ 8.56 (d, *J* = 2.3 Hz, 1H), 8.13 (dd, *J* = 8.7, 2.3 Hz, 1H), 7.82 − 7.71 (m, 2H), 7.60 (t, *J* = 7.4 Hz, 1H), 7.49 (t, *J* = 7.6 Hz, 2H), 7.14 (d, *J* = 8.7 Hz, 1H), 7.00 (s, 1H), 4.07 (s, 3H), 2.40 (s, 3H). [13]C NMR (101 MHz, Chloroform-*d*) δ 194.40, 193.37, 179.79, 163.34, 138.01, 137.13, 134.37, 132.89, 130.87, 129.89, 128.66, 120.09, 112.18, 102.68, 56.60, 25.08. [19]F NMR (376 MHz, Chloroform-*d*) δ -138.47 (20.9%), -138.53 (79.1%). [11]B NMR (128 MHz, Chloroform-*d*) δ -0.10. FT-IR (KBr, cm$^{-1}$): 3167.0, 3079.4, 2954.0, 2845.2, 1651.9, 1604.1, 1536.8, 1467.2, 1437.1, 1367.7, 1340.9, 1307.5, 1269.1, 1254.8, 1168.7, 1100.7, 1048.4, 1010.7, 978.5, 943.5, 876.5, 833.9, 798.8, 736.6, 708.9, 653.0, 632.6, 605.5, 567.2, 513.3, 471.0, 438.8. LRMS, m/z 345.1. HRMS (ESI) m/z found (calcd for $C_{18}H_{16}O_4^{10}BF_2$): 344.1137 (344.1141). The synthetic and purification procedures for compounds **2** and **3** are similar to those of compound **1**, which have been attached in Supplementary Information.

### Preparation of two-component afterglow materials by doping $BPBF_2$ compounds into organic matrices

For the preparation of $BPBF_2$-PhB-0.1% afterglow materials, 200 µL $BPBF_2$ in dichloromethane (1.0 mg/mL) and 200 mg phenylbenzoate (PhB) solids were added into an agate mortar (diameter = 5 cm). After solvent evaporation, the mixture of $BPBF_2$ and PhB was heated to 100 °C to form molten mixture. The molten mixture was allowed to stand at room temperature to give solidified melt-cast sample. The powder sample can obtained by grinding melt-cast sample into powder. Afterglow materials with different $BPBF_2$ dopants, different doping concentrations, different small organic matrices can be prepared by the procedure above.

## Data availability

The computational input and output files that support the findings of this study are available in Figshare with the identifier [https://doi.org/10.6084/m9.figshare.22292959.v4]. The X-ray crystallographic coordinates for structures of compound **1** reported in this study have been deposited at the Cambridge Crystallographic Data Centre (CCDC), under deposition numbers 2156303. These data can be obtained free of charge from The Cambridge Crystallographic Data Centre. The crystallographic data of compound **1** is also provided in the Supplementary Data 1. Other experimental data that support the findings of this study are available from the corresponding author upon request.

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

## Acknowledgements
We thank the financial supports from the National Natural Science Foundation of China (22175194, K. Z.), the Shanghai Scientific and Technological Innovation Project (20QA1411600, 20ZR1469200, K. Z.), and Hundred Talents Program from Shanghai Institute of Organic Chemistry (Y121078, K. Z.).

## Author contributions
J.L., X.L., G.W., X.W., M.W., and J.L. performed the experimental studies. J.L. and X.L. carried out the analysis. J.L., G.W. and X.W. performed the computational studies. K.Z. supervised the work.

## Competing interests
The authors declare no competing interests.
