## [Peer Review File · Nature Communications]

A Direct Observation of Up-Converted Room-Temperature Phosphorescence in an Anti-Kasha Dopant-Matrix SystemReviewers' comments:

Reviewer #1 (Remarks to the Author):

The authors reported a room-temperature phosphorescence (RTP) with phosphorescence max wavelength (λ_P) less than fluorescence (λ_F) in PhB doped BPBF2 and claimed that this phenomenon is due to the up-converted T2 emission, which violates Kasha's rule. However, this explanation of the observation is not valid since there are several flaws in their analysis.

(1) The main claim of the authors is that the second phosphorescence emission is from T2 \rightarrow S0, and the T2 is populated via thermally-activated ISC of S1 \rightarrow T2 and reverse internal conversion of T1 \rightarrow T2. However, this is not justified, and quantitative rates for these processes need to be carefully examined. The typical IC rate is 10^{-14} - 10^{-11} s, ISC rate is 10^{-8} - 10^{-3} s, and phosphorescence is 10^{-4} - 10^{-1} s. According to the detailed balance of thermal equilibrium between T1 and T2, if their energy gap is 0.33 eV (as in compound 2), then the rate constant $k(T2 \rightarrow T1) / k(T1 \rightarrow T2) = 377$ times (kT is only 0.026 eV at 300 K). Thus, the discussion regarding the energy gap on the T1 \rightarrow T2 rate constant is irrelevant, since the T2 \rightarrow T1 rate constant is about 400 times larger and the population of T1 over T2 is also 400 times larger! The only 0.25 % population of T2 won't give rise to such significant emissions.

(2) The way authors predict the rate was merely based on the coupling and the energy gap between two states, which is not believed to be even qualitatively correct. According to the classical Marcus theory, you need to consider the reorganization energy (E_r), and if the energy gap is larger than E_r , you will see a decrease in the rate. None of the discussion on the rate is based on legit theory. Even more, for photochemical processes, it is often required to consider the nuclear quantum effects and vibronic couplings, which can be captured by the quantum mechanical Fermi's golden rule (FGR) rate theory.

(3) An argument for T2 emission is that the SOC between T2 and S0 is large (about ten times larger than other pairs of states), but this won't be able to compete with such a large internal conversion rate from T2 to T1. If the authors could perform the FGR rate calculation for all IC, ISC, and phosphorescence processes, the competition between these channels will be clear.

(4) The observation seems to be a trivial effect, which could have originated from T1 phosphorescence from multiple molecular conformations. First, the quantum chemistry calculation doesn't consider finite temperature, so it is only based on a single optimized molecular geometry or conformation. But in reality, molecules are moving constantly and there may exist multiple minima that lead to emission mechanisms like some of the literature you cited. Could this be a twist-induced phenomenon? Could the authors rule out this possibility?

(5) Besides, the measurement is in the condensed phase, and the calculation is in a vacuum, so the "solvent" effect should be included to have a good estimate of the energy levels and couplings. B3LYP is widely used, but it is known to be inaccurate for many systems, so perhaps comparing it with other more advanced methods like RSH wB97X-D or even coupled-cluster methods is important to rule out the systematic error coming from B3LYP.

Thus, I recommend rejecting the current manuscript.

Reviewer #2 (Remarks to the Author):

I acknowledge that the authors report a significant finding, namely, the observation of up-converted room-temperature phosphorescence, which is anti-Kasha's rule, in BPBF2-doped PhB systems. A primary theoretical explanation has been forwarded and it is supported by the results of a series of intentionally designed experiments. In addition, some possible applications of BPBF2-doped PhB systems are shown (although I don't think they are essential). The results are interesting, novel and significant, and it's a very good work, but the story is still not good enough. As a consequent, I suggest to accept this manuscript after rational revisions.

Here are some questions and suggestions:

1. The authors have reasoned the significances of the direct observation of RTP(T_n) (RTP from T_n state, $n \geq 2$) in three aspects. The logic deduction is insufficient. Why is would be useful for the fabrication of visible-light-excitable RTP materials if the emitter has a smaller Stokes shift? Luminescent materials with Larger Stokes shift can also work well. In addition, for the third one, the authors wrote "since RTP(T_n) and RTP(T_1) possess different population mechanisms and very different k_{rel} values, some specific stimuli would have different influence on RTP(T_n) and RTP(T_1) emission intensities of the RTP materials". This is true, but this is applicable to all cases when there are differences in state population. Indeed, the authors observed the emission responses to the external stimuli of temperature and mechanical force. However, such kind of deduction is something like "afterglow" (be wise after the event). I suggest the authors reorganize this part.

2. There is a crucial point in this work. As claimed by the authors, "a direct observation of up-converted RTP(T_n) with $\lambda_P(T_n) < \lambda_F(S_1)$ would have significant impact on the straightforward and deep understanding of the behaviors of triplet excited states and the up-converted intersystem crossing". After a series of experiments and data analyses, what is (are) the straightforward and deep understanding? Just as the title of the manuscript "A Direct Observation ..."? An observation is not identical to an understanding. So, I strongly suggest the authors to think over the title and the discussion, to clearly present the essences of "the straightforward and deep understanding of the behaviors of triplet excited states and the up-converted intersystem crossing". For example, the factors

of “ $3n-\pi^*$ character” or/and proper ΔE_{T1T2} ? As for the title, “an Anti-Kasha Dopant-Matrix System” is a new phrase to me. I tried to find similar description in references, but I failed. The meaning of “an Anti-Kasha Dopant-Matrix System” is palpable, but it seems a little informal. This is unimportant, and the revision or modification is optional.

3. The discussion said that “The involvement of benzophenone functional groups on BPBF2 molecules is the key to achieve such up-converted RTP in the dopant-matrix systems since it not only facilitates ISC but also endows T_{sub} states with large k_P ”. It means that benzophenone moiety plays a key role in the system. In fact, there are many benzophenone derivatives, but the evident “Anti-Kasha Dopant-Matrix System” is solely observed for BPBF2-PhB system. This implies that BF2 is also a key role in the whole scenario. My question is: Is the possession of $3n-\pi^*$ character the real “boss” in the system? In addition, PhB also takes effect on the performance. I noticed, by reference browsing, PhB is commonly used as a matrix material in room-temperature phosphorescence (RTP) systems, I wonder if the matrix were replaced by other media, what would happen to the RTP experiments?

4. I don't agree with the authors on the description of “a serendipitous finding of up-converted RTP with $\lambda_P < \lambda_F$ and $\tau_P > 0.1$ s upon doping benzophenone-containing difluoroboron β -diketonate (BPBF2) into phenyl benzoate (PhB) matrices”. The observation of the delayed phosphorescence of $\lambda_P = 421$ nm is “a serendipitous finding”, but in the case of BPBF2-PhB system, there are careful designs, as indicated by the molecular synthesis and experiment setups.

5. The inset of Figure 5b should be clearly displayed, some digitals are super-positioned. The font size for Figure 4c can be reduced.

In summary, this work will attract broad interest from the researchers in relevant research fields for the novelty and the significance of the observations. The manuscript can be composed better to tell a more logical story, in order to be up to the standard of Nature Communications.

Reviewer #3 (Remarks to the Author): See Attachment

Organic afterglow emission is a very interesting phenomenon that has received a lot of attention at present. Anti-Kasha system will significantly help us understand the excited-states dynamics. The authors claim that the emission of high-energy region is originated from T2 state rather than TADF emission from RISC process, I think the evidences are not sufficient and credible unless the author can prove this with solid evidences rather than guess. Despite much work authors have done, there are still some parts remained to be well clarified. Considering the high requirements of this journal, this paper is therefore not suitable for publication in Nature communications. More detailed comments are shown below:

1. The authors think the high-energy region is relaxed from T2 state, and discusses in SI file. I can't accept the explanation. In this host-guest system, the peak of fluorescence spectra of compound in solution is very close to that of the higher-energy delayed emission band, why the high-energy emission do not arise from TADF emission of S1 of compound monomers? The authors also written that "Therefore, we compare the steady-state emission spectra and delayed emission spectra (1 ms delay) of 1-PhB-0.1% powders (both of which are collected in PhB matrices), as well as perform other experiments and TD-DFT calculations, for the assignment of emission bands." Where were the results of TD-DFT calculations? This manuscript didn't provide the energy level structures and the values of k_P , k_{ISC} , k_{RISC} , the SCOs from S_1 to T_n and so on. Therefore, I can't judge the rationality of the calculation results. The energy level positions obtained by emission spectra are unreasonable. Why the reverse internal conversion from T1 to T2 occurs under large bandgap? These should be elaborated by clear experimental results and computer calculations.
2. Why the intensities of high-energy peak decrease upon decreasing temperature? The authors interpret it as thermally activated ISC process. I still think it come from TADF emission from S1 due to the small bandgap between T2 and S1. Why do T2 and T1 emission have similar phosphorescent decay times, which seem to originate from the one excited state. Therefore, I suggest the more photophysical characteristics of pure compounds of 1-3 should be collected, include temperature-dependent delayed emission spectra and decay curves.
3. The temperature-dependent delayed emission spectra of the PhB compound should also be performed.
4. What was the critical progress compared with Previous manuscripts ((Angew. Chem. Int. Ed. 2021, 60, 17138; Adv. Funct. Mater. 2021, 2110207)?
5. The authors written that "However, to the best of our knowledge, in conventional conditions, such RTP(T_n) with $\lambda_P(T_n) < \lambda_F(S_1)$ have not been observed by experimental studies." However, as far as I know, the relative work has been reported (Angew. Chem. Int. Ed. 2020, 59, 10173-10178).
6. The authors written "and the SOCME value of S1 to T2 transition can reach 0.91 cm^{-1} ". The value is S1 to T2 or T2 to S1? Please check.
7. For compound 3, why phosphorescence intensity of T2 is higher than T1? If the 3 molecules are in monomeric form rather than in aggregation state in 3-PhB-0.1% samples, I strongly believe that high energy emission originates from TADF phenomenon rather than anti-Kasha rule.

Reviewers' comments:

Reviewer #1 (Remarks to the Author):

Overall Comment. The authors reported a room-temperature phosphorescence (RTP) with phosphorescence max wavelength (λ_P) less than fluorescence (λ_F) in PhB doped BPBF2 and claimed that this phenomenon is due to the up-converted T2 emission, which violates Kasha's rule. However, this explanation of the observation is not valid since there are several flaws in their analysis.

Response: We sincerely thank the reviewer for the professional comments and constructive suggestion. In this revised manuscript, we have corrected the mistake. Since the detailed comments are listed in the following, we make corresponding response in the following.

Comment 1. The main claim of the authors is that the second phosphorescence emission is from T2 \rightarrow S0, and the T2 is populated via thermally-activated ISC of S1 \rightarrow T2 and reverse internal conversion of T1 \rightarrow T2. However, this is not justified, and quantitative rates for these processes need to be carefully examined. The typical IC rate is 10^{-14} - 10^{-11} s, ISC rate is 10^{-8} - 10^{-3} s, and phosphorescence is 10^{-4} - 10^{-1} s. According to the detailed balance of thermal equilibrium between T1 and T2, if their energy gap is 0.33 eV (as in compound 2), then the rate constant $k(T2 \rightarrow T1) / k(T1 \rightarrow T2) = 377$ times (kT is only 0.026 eV at 300 K). Thus, the discussion regarding the energy gap on the T1 \rightarrow T2 rate constant is irrelevant, since the T2 \rightarrow T1 rate constant is about 400 times larger and the population of T1 over T2 is also 400 times larger! The only 0.25 % population of T2 won't give rise to such significant emissions.

Response 1: We thank the reviewer for the valuable comments and suggestion. The large $k_P(T2) / k_P(T1)$ can counterbalance the small population of T2 states, leading to the emergence of RTP(T2) in the delayed emission spectra. For example, in the case of compound 2, the T2 state of compound 2 exhibits significant n- π^* transition character with large SOCME of 26.09 cm^{-1} and transition dipole moment of 7.62×10^{-3} a.u.

calculated at TD- ω B97X-D3/def2-tzvp(-f) level of theory (Table 5). In the literature, Kaji and coworkers reported the theoretical calculation of quantitative rates of the photophysical processes in benzophenone systems (ref. 44 and 45 in the main text). Here the luminescent compounds contain benzophenone functional groups, so we use Kaji's method to calculate the rate constants of phosphorescence decay (computational details have been attached in the revised supporting information). The $k_P(T_2)$ at optimized T_2 geometry and $k_P(T_1)$ at optimized T_1 geometry of compound **2** are calculated to $9.28 \times 10^3 \text{ s}^{-1}$ and $1.06 \times 10^1 \text{ s}^{-1}$, respectively, giving $k_P(T_2)/k_P(T_1)$ ratio close to 10^3 (Table 5). Recent studies of anti-Kasha systems (ref. 48-50 in the main text) showed that electron-vibrational coupling should be taken into account for the calculation of both $k(T_n-T_1)$ and $k(T_1-T_n)$. Accordingly, based on the FCclasses software (ref. 51 in the main text), $k(T_n-T_1)$ and $k(T_1-T_n)$ of compound **2** have been calculated to be $2.67 \times 10^{11} \text{ s}^{-1}$ and $7.30 \times 10^8 \text{ s}^{-1}$, respectively (Table 4 and Table S30). The $k(T_1-T_n)/k(T_n-T_1)$ ratios have been calculated to be 2.73×10^{-3} at 300 K (Table 4). The rate constants of forward and reverse internal conversion were calculated based on frequency analysis at the ω B97XD/6-31G(d,p) level of theory and TD-DFT calculated results at the ω B97X-D3/def2-TZVP(-f) level of theory. Given that the relative emission intensity of RTP(T_2)/RTP(T_1) is proportional to $k(T_1-T_2)/k(T_2-T_1) \times k_P(T_2)/k_P(T_1)$, the above theoretical calculations support the experimental observation of RTP(T_2)/RTP(T_1) dual emission in the delayed spectra (1 ms delay). Corresponding revisions, as well as the results of compound **1** and **3**, have been added in this revised manuscript.

Comment 2. The way authors predict the rate was merely based on the coupling and the energy gap between two states, which is not believed to be even qualitatively correct. According to the classical Marcus theory, you need to consider the reorganization energy (E_r), and if the energy gap is larger than E_r , you will see a decrease in the rate. None of the discussion on the rate is based on legit theory. Even more, for photochemical processes, it is often required to consider the nuclear quantum effects and vibronic couplings, which can be captured by the quantum mechanical Fermi's

golden rule (FGR) rate theory.

Response 2: We thank the reviewer for the valuable comments and suggestion. Accordingly, we use the quantum mechanical Fermi's golden rule rate theory to calculate the rate constants of intersystem crossing, internal conversion and radiative decay in this revised manuscript. In the literature, Kaji and coworkers reported the theoretical calculation of quantitative rates of the photophysical processes in benzophenone systems (ref. 44 and 45 in the main text). Here the luminescent compounds contain benzophenone functional groups, so we use Kaji's method to calculate the rate constants of intersystem crossing, internal conversion and radiative decay (computational details have been attached in the revised supporting information). In addition, Marcus theory has also been used to calculate the rate constants of intersystem crossing for the present system. Both FGR and Marcus theory show that the ISC rate constants of S_1 -to- T_1 and S_1 -to- T_n ($n = 2$ for compounds **1** and **2**, $n = 3$ for compound **3**) are above 10^7 s⁻¹ (Table 3). Given the fluorescence decay of S_1 states of intramolecular charge transfer nature has rate constants of 10^7 to 10^8 s⁻¹ (Table 3), such large ISC rate constants would result in relatively high ISC quantum yields in the system. Actually, in the experimental studies, the steady-state emission spectra of **1-3** solutions in dichloromethane at 77 K exhibit significant components of phosphorescence signals (Figure S15), which also support the strong tendency of ISC in the present system.

Recent studies of anti-Kasha systems (ref. 48-50 in the main text) showed that electron-vibrational coupling should be taken into account for the calculation of both $k(T_n-T_1)$ and $k(T_1-T_n)$. Accordingly, based on the FCclasses software (ref. 51 in the main text), $k(T_n-T_1)$ and $k(T_1-T_n)$ have been calculated to be $10^{10}\sim 10^{11}$ s⁻¹ and $10^8\sim 10^9$ s⁻¹, respectively (Table 4 and Table S30). The $k(T_1-T_n)/k(T_n-T_1)$ ratios have been calculated to be on the order of $10^{-3}\sim 10^{-2}$ at 300 K (Table 4). In our previous manuscript, the estimation of $k(T_1-T_2)$ values (to be on the order of $10^0\sim 10^1$ s⁻¹) is not correct. We have corrected the mistake of $k(T_1-T_2)$ in this revised manuscript.

Phosphorescence rate constants, $k_P(T_1)$ and $k_P(T_n)$, have been obtained by FGR rate theory (Table 5), using Kaji's method at TD- ω B97X-D3/def2-tzvp(-f) level of

theory. The $k_P(T_n)$ values of $10^3\sim 10^4\text{ s}^{-1}$ at T_n geometries ($n = 2$ for compounds **1** and **2**, $n = 3$ for compound **3**) are found to be much larger than $k_P(T_1)$ values of $10^0\sim 10^1\text{ s}^{-1}$ at T_1 geometries, exhibiting $k_P(T_n)/k_P(T_1)$ ratios of $10^2\sim 10^3$. Given that the relative emission intensity of $\text{RTP}(T_n)/\text{RTP}(T_1)$ is proportional to $k(T_1-T_n)/k(T_n-T_1) \times k_P(T_n)/k_P(T_1)$, the above theoretical calculations support the experimental observation of $\text{RTP}(T_n)/\text{RTP}(T_1)$ dual emission in the delayed spectra.

Overall, the calculation results agree with the experimental observation of $\text{RTP}(T_n)/\text{RTP}(T_1)$ dual emission in the present study. Please also find the details in the response to Reviewer 1's Comment 3. Corresponding revisions have been made in this revised manuscript.

Comment 3. An argument for T2 emission is that the SOC between T2 and S0 is large (about ten times larger than other pairs of states), but this won't be able to compete with such a large internal conversion rate from T2 to T1. If the authors could perform the FGR rate calculation for all IC, ISC, and phosphorescence processes, the competition between these channels will be clear.

Response 3: We thank the reviewer for the valuable comments and suggestion. Accordingly, we perform the FGR rate calculation for intersystem crossing, internal conversion, and phosphorescence decay in the present system. In the literature, Kaji and coworkers reported the theoretical calculation of quantitative rates of the photophysical processes in benzophenone systems (ref. 44 and 45 in the main text). Here the luminescent compounds contain benzophenone functional groups, so we use Kaji's method to calculate the rate constants of the photophysical processes (computational details have been attached in the revised supporting information). The excited state energy levels of compounds **1** to **3** have been calculated by TD- ω B97X-D3/def2-tzvp(-f) method for the S_0 , S_1 , T_1 and T_n geometries of compounds **1** to **3**; the geometries of ground states and excited states were optimized at ω B97XD/6-31G(d, p) level of theory. Frequency analysis was performed at the same theoretical level to validate the presence of minimum and to generate the hessian matrix, which is needed for the rate calculation.

For the forward ISC, SOCME values of S_1 to T_n for different geometries have been calculated on ORCA 5.0.3 program with spin-orbit mean-field (SOMF) methods at ω B97X-D3/def2-tzvp(-f) level. Table 3 and Table S15-S17 show that most ISC channels have SOCME above 1 cm^{-1} and some ISC channels possess SOCME larger than 10 cm^{-1} , which suggest the strong tendency of forward ISC in the system. The FGR rate theory has been applied to calculate ISC rate constants as shown in Table 3 and Table S24-S26; the ISC rate constants are calculated based on Kaji's method (detailed in Supporting Information). Besides, the calculations of ISC rate constants via Marcus theory have also been performed (ref. 46 and 47 in the main text, Table 3 and Table S27-S29). Both FGR and Marcus theory show that the ISC rate constants of S_1 -to- T_1 and S_1 -to- T_n ($n = 2$ for compounds **1** and **2**, $n = 3$ for compound **3**) are above 10^7 s^{-1} (Table 3). Given the fluorescence decay of S_1 states of intramolecular charge transfer nature has rate constants of 10^7 to 10^8 s^{-1} (Table 3), such large ISC rate constants would result in relatively high ISC quantum yields in the system. Actually, in the experimental studies, the steady-state emission spectra of **1-3** solutions in dichloromethane at 77 K exhibit significant components of phosphorescence signals (Figure S15), which also support the strong tendency of ISC in the present system. For the reverse ISC, from the results obtained by both B3LYP and ω B97X-D3 methods, it is found that, at the optimized geometry of either T_1 or T_n ($n = 2$ for compounds **1** and **2**, $n = 3$ for compound **3**), the T_1 and T_n levels are much lower than S_1 levels (Table 2). Given that reverse ISC starts from triplet excited states, these results suggest that reverse ISC is not likely to occur. The corresponding rate constants of reverse ISC have also been calculated to show small values (Table S24-S29), which can explain the absence of TADF afterglow in the experimental observations.

Upon forward ISC, we propose that, based on the experimental observations summarized in Table 1 and shown in Figure 3c, the resultant T_1 and T_n states would build T_1 - T_n equilibrium via forward and reverse internal conversion. Kaji's method has been used to calculate the rate constants of forward internal conversion; frequency analyses are performed at ω B97XD/6-31G(d,p) level of theory (detailed in Supporting Information). It is found that forward internal conversion is very fast with rate constant

of $k(T_n-T_1)$ on the order of $10^9\sim 10^{13} \text{ s}^{-1}$ (Table 4). For the reverse internal conversion, the rate constants $k(T_1-T_n)$ calculated by the Arrhenius-type expression (Supporting Information) are found to be largely underestimated (Table 4). Recent studies of anti-Kasha systems (ref. 48-50 in the main text) showed that electron-vibrational coupling should be taken into account for the calculation of both $k(T_n-T_1)$ and $k(T_1-T_n)$. Accordingly, based on the FCclasses software (ref. 51 in the main text), $k(T_n-T_1)$ and $k(T_1-T_n)$ have been calculated to be $10^{10}\sim 10^{11} \text{ s}^{-1}$ and $10^8\sim 10^9 \text{ s}^{-1}$, respectively (Table 4 and Table S30). The $k(T_1-T_n)/k(T_n-T_1)$ ratios have been calculated to be on the order of $10^{-3}\sim 10^{-2}$ at 300 K (Table 4), much larger than those estimated by the Arrhenius-type expression (Table 4). To investigate the phosphorescence decay of T_1 and T_n states, the corresponding SOCME values, transition dipole moments, and phosphorescence rate constants have been calculated and summarized in Table 5. Calculated at TD- ω B97X-D3/def2-tzvp(-f) level of theory, the SOCME values and transition dipole moments of T_n-S_0 phosphorescence decay at T_n geometries ($n = 2$ for compounds **1** and **2**, $n = 3$ for compound **3**) have been found to be much larger than those of T_1-S_0 phosphorescence decay at T_1 geometries (Table 5). Phosphorescence rate constants, $k_P(T_1)$ and $k_P(T_n)$, have been obtained by FGR rate theory (Table 5). The $k_P(T_n)$ values of $10^3\sim 10^4 \text{ s}^{-1}$ at T_n geometries ($n = 2$ for compounds **1** and **2**, $n = 3$ for compound **3**) are found to be much larger than $k_P(T_1)$ values of $10^0\sim 10^1 \text{ s}^{-1}$ at T_1 geometries, exhibiting $k_P(T_n)/k_P(T_1)$ ratios of $10^2\sim 10^3$. Given that the relative emission intensity of $\text{RTP}(T_n)/\text{RTP}(T_1)$ is proportional to $k(T_1-T_n)/k(T_n-T_1) \times k_P(T_n)/k_P(T_1)$, the above theoretical calculations support the experimental observation of $\text{RTP}(T_n)/\text{RTP}(T_1)$ dual emission in the delayed spectra.

Based on the experimental observation and theoretical calculation, Figure 4a illustrates the photophysical mechanism of the dual RTP systems. Upon excitation, S_1 states of intramolecular charge transfer character form. Because of the involvement of benzophenone functional groups, different symmetry between S_1 states and triplet excited states, and relatively small singlet-triplet splitting energies, the system shows strong tendency to undergo intersystem crossing to form T_1 and T_n states. Upon intersystem crossing, the T_1 and T_n states build equilibrium under ambient conditions

due to the fast internal conversion and reverse internal conversion facilitated by electron-vibrational coupling. The T_n states of $n-\pi^*$ characters from benzophenone groups ($n = 2$ for compounds **1** and **2**, $n = 3$ for compound **3**) have large phosphorescence decay rates and counterbalance the small population of T_n states, leading to RTP(T_n) emission that violate Kasha's rule. The PhB matrices can provide rigid microenvironment to suppress nonradiative decay (k_{nr}) and oxygen quenching (k_q) of BPBF₂'s triplet excited states. The T_1 states have small phosphorescence rates of $10^0 \sim 10^1 \text{ s}^{-1}$ to show afterglow property with lifetimes of several hundred milliseconds under ambient conditions. Because of the long-lived excited state nature, the T_1 states also serve as store for the population of T_n states via T_1 - T_n equilibrium, which is the reason that RTP(T_n) also possess lifetimes of several hundred milliseconds.

All the above results and discussion have been added in this revised manuscript.

Comment 4. The observation seems to be a trivial effect, which could have originated from T_1 phosphorescence from multiple molecular conformations. First, the quantum chemistry calculation doesn't consider finite temperature, so it is only based on a single optimized molecular geometry or conformation. But in reality, molecules are moving constantly and there may exist multiple minima that lead to emission mechanisms like some of the literature you cited. Could this be a twist-induced phenomenon? Could the authors rule out this possibility?

Response 4: We thank the reviewer for the valuable comments and suggestion. We understand that T_1 states with different conformations give phosphorescence bands with different emission maxima. In the reported studies (*J. Mater. Chem. C*, **2018**, *6*, 9238), the axial and equatorial conformation of the T_1 state of phenothiazine-containing compound at room temperature have been found to exhibit higher-energy (local minimum) and lower-energy (global minimum) phosphorescence bands, respectively. At low temperature, the lower-energy bands decrease while the higher-energy bands still exist (*J. Mater. Chem. C*, **2018**, *6*, 9238). One may reason that the 421 nm and 483 nm bands in the present study originate from local minimum and global minimum of **1**'s T_1 states, respectively. If this is true, the 421 nm higher-energy band should still

exist at low temperature, but the variable temperature delayed emission spectra of **1**-PhB materials show the decrease and absence of 421 nm band upon lowering temperature (Figure 1f). Therefore, the conformation-dependent or twist-induced T_1 level change is not likely to be the case in the present system with dual RTP property. Besides, in a very recent study of BF₂bdk-matrix system reported by our group (*Adv. Opt. Mater.* **2022**, 202202540), the twisted BF₂bdk compound showed RTP spectral shift upon conformation change, whereas dual RTP has not been observed. In addition, compound **1** has only one conformation in single crystal structure (Figure S40). Moreover, TD-DFT calculations have also been performed to investigate the dependence of T_1 levels on **1**'s conformation (Figure S8); the conformation is defined by the twisted angle between aromatic donor and dioxaborine acceptor. The potential energy surface of **1**'s T_1 state as a function of the twisted angle shows only one energy minimum (Figure S8). These results and analyses exclude the possibility of twist-induced dual RTP in the present system.

Comment 5. Besides, the measurement is in the condensed phase, and the calculation is in a vacuum, so the "solvent" effect should be included to have a good estimate of the energy levels and couplings. B3LYP is widely used, but it is known to be inaccurate for many systems, so perhaps comparing it with other more advanced methods like RSH wB97X-D or even coupled-cluster methods is important to rule out the systematic error coming from B3LYP.

Response 5: We thank the reviewer for the valuable comments and suggestion. Accordingly, in this revised manuscript, the calculations of excited state energy levels at TD-B3LYP/G/def2-TZVP(-f) level of theory (without considering conductor-like polarizable continuum model) (Table S2-S4), at TD-B3LYP/G/def2-TZVP(-f) level of theory (considering conductor-like polarizable continuum model (CH₂Cl₂)) (Table S5-S7), and at TD- ω B97X-D3/def2-TZVP(-f) level of theory (Table S8-S10) have been performed. It is found that the energy levels obtained by TD- ω B97X-D3/def2-TZVP(-f) calculation are relatively close to those by experimental observations (Table S11), so we use the results obtained by TD- ω B97X-D3/def2-TZVP(-f) method for the FGR rate

calculation in the present study.

Thus, I recommend rejecting the current manuscript.

Reviewer #2 (Remarks to the Author):

Comments on of NCOMMS-22-12909A-Z

I acknowledge that the authors report a significant finding, namely, the observation of up-converted room-temperature phosphorescence, which is anti-Kasha's rule, in BPBF2-doped PhB systems. A primary theoretical explanation has been forwarded and it is supported by the results of a series of intentionally designed experiments. In addition, some possible applications of BPBF2-doped PhB systems are shown (although I don't think they are essential). The results are interesting, novel and significant, and it's a very good work, but the story is still not good enough. As a consequent, I suggest to accept this manuscript after rational revisions.

Response: We sincerely thank the reviewer for the strong endorsement on the interesting nature and high quality of our work. Since the reviewer made detailed comments and suggestion in the following, we make corresponding responses in the following.

Here are some questions and suggestions:

Comment 1. The authors have reasoned the significances of the direct observation of RTP(T_n) (RTP from T_n state, $n \geq 2$) in three aspects. The logic deduction is insufficient. Why is would be useful for the fabrication of visible-light-excitable RTP materials if the emitter has a smaller Stokes shift? Luminescent materials with Larger Stokes shift can also work well. In addition, for the third one, the authors wrote "since RTP(T_n) and RTP(T_1) possess different population mechanisms and very different k_P values, some specific stimuli would have different influence on RTP(T_n) and RTP(T_1) emission intensities of the RTP materials". This is true, but this is applicable to all cases when there are differences in state population. Indeed, the authors observed the emission

responses to the external stimuli of temperature and mechanical force. However, such kind of deduction is something like “afterglow” (be wise after the event). I suggest the authors reorganize this part.

Response 1: We thank the reviewer for the valuable comment and suggestion.

For the part of “smaller Stokes shift”, the corresponding sentences have been revised as follows. “RTP(T_n) exhibits smaller Stokes shift than RTP(T_1), which would be useful for the fabrication of visible-light-excitable deep-blue RTP materials. To be fair, luminescent materials with large Stokes shift can minimize the interference of scattered light from excitation source; this is an advantage of conventional RTP materials. However, in the case of deep-blue RTP materials, large Stokes shift means that high-energy UV sources (which may destabilize organic materials) are required to excite the materials; for instance, in the reported studies, UV lights of short wavelengths such as 310 nm, 280 nm or even shorter are used to switch on the deep-blue RTP property^{5,9}. RTP(T_n) with small Stokes shift would provide a new pathway to achieve deep-blue RTP materials that can be excited by visible light or UVA light. Because of the long-lived excited state nature of RTP materials, the interference from excitation source and background fluorescence can be eliminated by time-gated or afterglow mode.”

For the part of “stimuli-responsive RTP materials”, we have rewritten the sentences as follows. “Third, the involvement of RTP(T_n) would endow organic systems with RTP(T_1) plus RTP(T_n) dual phosphorescence property. Given that RTP(T_n) and RTP(T_1) possess different population mechanisms and very different phosphorescence decay rates, if some specific stimuli have different influence on RTP(T_n) and RTP(T_1) emission intensities, the organic systems would give significant RTP(T_n)/RTP(T_1) ratiometric response to function as stimuli-responsive RTP materials.”

Comment 2. There is a crucial point in this work. As claimed by the authors, “a direct observation of up-converted RTP(T_n) with $\lambda_P(T_n) < \lambda_F(S1)$ would have significant impact on the straightforward and deep understanding of the behaviors of triplet excited states and the up-converted intersystem crossing”. After a series of experiments and

data analyses, what is (are) the straightforward and deep understanding? Just as the title of the manuscript “A Direct Observation ...”? An observation is not identical to an understanding. So, I strongly suggest the authors to think over the title and the discussion, to clearly present the essences of “the straightforward and deep understanding of the behaviors of triplet excited states and the up-converted intersystem crossing”. For example, the factors of “ $^3n-\pi^*$ character” or/and proper $\Delta E(T_1T_2)$? As for the title, “an Anti-Kasha Dopant-Matrix System” is a new phrase to me. I tried to find similar description in references, but I failed. The meaning of “an Anti-Kasha Dopant-Matrix System” is palpable, but it seems a little informal. This is unimportant, and the revision or modification is optional.

Response 2: We thank the reviewer for the valuable comment and suggestion. In this revised manuscript, we summarize the understanding of behaviors of higher triplet excited states in the last part of the manuscript. The involvement of benzophenone functional groups on BPBF₂ molecules is very important to achieve such up-converted RTP in the dopant-matrix systems since it not only facilitates ISC but also endows T_n ($n \geq 2$) states with $n-\pi^*$ character and large k_p . Given that the energy levels of the T_n states are mainly determined by the benzophenone groups, here the use of difluoroboron β -diketonate functional groups (with suitable LUMO level and electron-accepting strength) is also very important to result in a proper $\Delta E(T_n-T_1)$ in BPBF₂ system. With the above understanding, we select anthraquinone functional groups whose $^3n-\pi^*$ states also possess large k_p to prepare luminescent molecules. When some specific electron-donating functional groups are linked to anthraquinone groups to give proper $\Delta E(T_n-T_1)$, preliminary studies show the observation of RTP(T_n) bands of typical $^3n-\pi^*$ character from anthraquinone functional groups (data not shown).

The present study shows that it is still possible to form T₁-T_n equilibrium under ambient conditions in organic systems with $\Delta E(T_n-T_1)$ of around 0.3 eV. Theoretical studies reveal that the electron-vibrational coupling can increase the population of T_n states, and the large $k_p(T_n)/k_p(T_1)$ ratios can compensate the small population of T_n states, leading to anti-Kasha RTP(T_n) emission.

The change of RTP(T_n) emission reflects the change of photophysical processes

related to T_n states, so the direct observation of RTP(T_n) facilitates the study of the population, equilibrium, and radiative decay of T_n states. The present study would have significant impact on the deep understanding of photophysical behaviors of higher triplet excited states and provide new strategies for designing high-performance organic afterglow materials with intriguing properties.

Comment 3. The discussion said that “The involvement of benzophenone functional groups on BPBF₂ molecules is the key to achieve such up-converted RTP in the dopant-matrix systems since it not only facilitates ISC but also endows T₂ states with large k_P ”. It means that benzophenone moiety plays a key role in the system. In fact, there are many benzophenone derivatives, but the evident “Anti-Kasha Dopant-Matrix System” is solely observed for BPBF₂-PhB system. This implies that BF₂ is also a key role in the whole scenario. My question is: Is the possession of $3n-\pi^*$ character the real “boss” in the system? In addition, PhB also takes effect on the performance. I noticed, by reference browsing, PhB is commonly used as a matrix material in room-temperature phosphorescence (RTP) systems, I wonder if the matrix were replaced by other media, what would happen to the RTP experiments?

Response 3: We thank the reviewer for the valuable comment and suggestion. The involvement of benzophenone functional groups on BPBF₂ molecules is very important to achieve such up-converted RTP in the dopant-matrix systems since it not only facilitates ISC but also endows T_n ($n \geq 2$) states with $n-\pi^*$ character and large k_P . Given that the energy levels of the T_n states are mainly determined by the benzophenone groups, here the use of difluoroboron β -diketonate functional groups (with suitable LUMO level and electron-accepting strength) is also very important to result in a proper $\Delta E(T_n-T_1)$ in BPBF₂ system.

We use dopant-matrix design strategy to construct organic afterglow materials, where the selection of organic matrix is very important. The selection guideline of organic matrix is based on its role in BF₂bdk-matrix afterglow system³⁷, where BF₂bdk represents difluoroboron β -diketonate compound. (a) Organic matrix should suppress nonradiative decay and oxygen quenching of BF₂bdk's T_1 states, so that crystalline

matrix is preferred. (b) In BF₂bdk-matrix system, organic matrices with carbonyl or ester groups interact with BF₂bdk's S₁ states via dipole-dipole interactions, lower BF₂bdk's S₁ levels (BF₂bdk's T₁ levels are less influenced by matrix's environment), and thus reduce ΔE_{ST} and facilitate intersystem crossing³⁸. This dipole effect in enhancing intersystem crossing has also been proved in a recent reported study³⁹. Here phenyl benzoate (PhB) and benzophenone (BP) are used to accommodate BPBF₂ because of their crystalline natures and relatively large dipole moments in the ground states; PhB and BP are two of the most frequently used matrices developed in our lab. By doping 0.1 wt% BPBF₂ into BP (BP has ground-state dipole moments of 2.96 D as estimated by TD-B3LYP/6-31G(d,p)), the resultant dopant-matrix samples have been found to show insignificant afterglow at room temperature (Figure S3); BP matrix has relatively low T₁ level (2.76 eV, estimated from phosphorescence maxima) to receive excited state energy from BPBF₂'s T₁ states, causing the quenching of organic afterglow in BPBF₂-BP samples^{40,41}. Cyclo olefin polymer (COP) with high T₁ level but insignificant dipole moment has also been test as organic matrix. The BPBF₂-COP samples show insignificant room-temperature afterglow (Figure S4). PhB has ground-state dipole moments of 1.94 D and a high T₁ level of 3.53 eV as calculated by TD-B3LYP/6-31G(d,p). Upon doping BPBF₂ into PhB, the resultant BPBF₂-PhB materials exhibit significant room-temperature afterglow.

These results and descriptions have been added in this revised manuscript.

Comment 4. I don't agree with the authors on the description of "a serendipitous finding of upconverted RTP with $\lambda_P < \lambda_F$ and $\tau_P > 0.1$ s upon doping benzophenone-containing difluoroboron β -diketonate (BPBF₂) into phenyl benzoate (PhB) matrices". The observation of the delayed phosphorescence of $\lambda_P = 421$ nm is "a serendipitous finding", but in the case of BPBF₂-PhB system, there are careful designs, as indicated by the molecular synthesis and experiment setups.

Response 4: We thank the reviewer for the valuable comment and suggestion. We make corresponding revisions in page 6 of this revised manuscript. "Here we report a serendipitous finding of up-converted RTP with $\lambda_P < \lambda_F$ and $\tau_P > 0.1$ s upon doping

benzophenone-containing difluoroboron β -diketonate (BPBF₂) into phenyl benzoate (PhB) matrices. The BPBF₂-PhB materials are prepared by rational material design based on dopant-matrix strategy, while the up-converted RTP is from an unexpected observation.”

Comment 5. The inset of Figure 5b should be clearly displayed, some digitals are super-positioned. The font size for Figure 4c can be reduced. In summary, this work will attract broad interest from the researchers in relevant research fields for the novelty and the significance of the observations. The manuscript can be composed better to tell a more logical story, in order to be up to the standard of Nature Communications.

Response 5: We thank the reviewer for the careful review. The inset of Figure 5b and the font size of Figure 4c have been revised. We sincerely thank the reviewer for the professional comments and strong endorsement on our manuscript.

Reviewer #3 (Remarks to the Author):

Overall Comment. Organic afterglow emission is a very interesting phenomenon that has received a lot of attention at present. Anti-Kasha system will significantly help us understand the excited-states dynamics. The authors claim that the emission of high-energy region is originated from T₂ state rather than TADF emission from RISC process, I think the evidences are not sufficient and credible unless the author can prove this with solid evidences rather than guess. Despite much work authors have done, there are still some parts remained to be well clarified. Considering the high requirements of this journal, this paper is therefore not suitable for publication in Nature communications.

Response: We thank the reviewer for the valuable comments and suggestion. In this revised manuscript, we provide solid evidence to rule out TADF mechanism from both experimental studies and theoretical calculations, and support the proposed mechanism of up-converted room-temperature phosphorescence. Please find the response to reviewer’s detailed comments in the following.

More detailed comments are shown below:

Comment 1. The authors think the high-energy region is relaxed from T2 state, and discusses in SI file. I can't accept the explanation. In this host-guest system, the peak of fluorescence spectra of compound in solution is very close to that of the higher-energy delayed emission band, why the high-energy emission do not arise from TADF emission of S1 of compound monomers? The authors also written that "Therefore, we compare the steady-state emission spectra and delayed emission spectra (1 ms delay) of 1-PhB-0.1% powders (both of which are collected in PhB matrices), as well as perform other experiments and TD-DFT calculations, for the assignment of emission bands." Where were the results of TD-DFT calculations? This manuscript didn't provide the energy level structures and the values of KP, kISC, KRISC, the SCOs from S1 to Tn and so on. Therefore, I can't judge the rationality of the calculation results. The energy level positions obtained by emission spectra are unreasonable. Why the reverse internal conversion from T1 to T2 occurs under large bandgap? These should be elaborated by clear experimental results and computer calculations.

Response 1: We thank the reviewer for the valuable comment and suggestion. It is known that microenvironment has significant influence on the position of fluorescence peak of luminescent compounds with intramolecular charge transfer character. Although the peak of fluorescence spectra of compound **1** in dichloromethane solution (423 nm, Table S1) is very close to that of the higher-energy delayed emission band (1 ms delay) of **1**-PhB-0.1% powders (421 nm, Figure 1c), the 421 nm delayed emission of **1**-PhB-0.1% powders *cannot* simply be attributed to TADF because the microenvironment of compound **1** in dichloromethane solution is *different* from that in PhB matrices. We hope the reviewer can understand that it is fair that we compare the steady-state emission spectra and delayed emission spectra (1 ms delay) of **1**-PhB-0.1% powders (*both of which are collected in PhB matrices*), when we assign the 421 delayed emission.

In this revised manuscript, the excited state energy levels, electron-hole isosurface map of excited states, SOCME values from S₁ to T_n have been attached in supporting information (Figure S9, S11 and S13). In addition, we perform the FGR rate calculation for intersystem crossing, internal conversion, and phosphorescence decay in the present

system. In the literature, Kaji and coworkers reported the theoretical calculation of quantitative rates of the photophysical processes in benzophenone systems^{44,45}. Here the luminescent compounds contain benzophenone functional groups, so we use Kaji's method to calculate the rate constants of the photophysical processes (computational details have been attached in the revised supporting information). The excited state energy levels of compounds **1** to **3** have been calculated by TD- ω B97X-D3/def2-tzvp(-f) method for the S₀, S₁, T₁ and T_n geometries of compounds **1** to **3**; the geometries of ground states and excited states were optimized at ω B97XD/6-31G(d, p) level of theory. Frequency analysis was performed at the same theoretical level to validate the presence of minimum and to generate the hessian matrix, which is needed for the rate calculation.

For the forward ISC, SOCME values of S₁ to T_n for different geometries have been calculated on ORCA 5.0.3 program with spin-orbit mean-field (SOMF) methods at ω B97X-D3/def2-tzvp(-f) level. Table 3 and Table S15-S17 show that most ISC channels have SOCME above 1 cm⁻¹ and some ISC channels possess SOCME larger than 10 cm⁻¹, which suggest the strong tendency of forward ISC in the system. The FGR rate theory has been applied to calculate ISC rate constants as shown in Table 3 and Table S24-S26; the ISC rate constants are calculated based on Kaji's method^{44,45} (detailed in Supporting Information). Besides, the calculations of ISC rate constants via Marcus theory have also been performed^{46,47} (Table 3 and Table S27-S29). Both FGR and Marcus theory show that the ISC rate constants of S₁-to-T₁ and S₁-to-T_n (n = 2 for compounds **1** and **2**, n = 3 for compound **3**) are above 10⁷ s⁻¹ (Table 3). Given the fluorescence decay of S₁ states of intramolecular charge transfer nature has rate constants of 10⁷ to 10⁸ s⁻¹ (Table 3), such large ISC rate constants would result in relatively high ISC quantum yields in the system. Actually, in the experimental studies, the steady-state emission spectra of **1-3** solutions in dichloromethane at 77 K exhibit significant components of phosphorescence signals (Figure S15), which also support the strong tendency of ISC in the present system. For the reverse ISC, from the results obtained by both B3LYP and ω B97X-D3 methods, it is found that, at the optimized geometry of either T₁ or T_n (n = 2 for compounds **1** and **2**, n = 3 for compound **3**), the T₁ and T_n levels are much lower than S₁ levels (Table 2). Given that reverse ISC starts

from triplet excited states, these results suggest that reverse ISC is not likely to occur. The corresponding rate constants of reverse ISC have also been calculated to show small values (Table S24-S29), which can explain the absence of TADF afterglow in the experimental observations.

Upon forward ISC, we propose that, based on the experimental observations summarized in Table 1 and shown in Figure 3c, the resultant T_1 and T_n states would build T_1 - T_n equilibrium via forward and reverse internal conversion. Kaji's method^{44,45} has been used to calculate the rate constants of forward internal conversion; frequency analyses are performed at ω B97XD/6-31G(d,p) level of theory (detailed in Supporting Information). It is found that forward internal conversion is very fast with rate constant of $k(T_n-T_1)$ on the order of $10^9\sim 10^{13} \text{ s}^{-1}$ (Table 4). For the reverse internal conversion, the rate constants $k(T_1-T_n)$ calculated by the Arrhenius-type expression (Supporting Information) are found to be largely underestimated (Table 4). Recent studies of anti-Kasha systems⁴⁸⁻⁵⁰ showed that electron-vibrational coupling should be taken into account for the calculation of both $k(T_n-T_1)$ and $k(T_1-T_n)$. Accordingly, based on the FCclasses software⁵¹, $k(T_n-T_1)$ and $k(T_1-T_n)$ have been calculated to be $10^{10}\sim 10^{11} \text{ s}^{-1}$ and $10^8\sim 10^9 \text{ s}^{-1}$, respectively (Table 4 and Table S30). The $k(T_1-T_n)/k(T_n-T_1)$ ratios have been calculated to be on the order of $10^{-3}\sim 10^{-2}$ at 300 K (Table 4), much larger than those estimated by the Arrhenius-type expression (Table 4). To investigate the phosphorescence decay of T_1 and T_n states, the corresponding SOCME values, transition dipole moments, and phosphorescence rate constants have been calculated and summarized in Table 5. Calculated at TD- ω B97X-D3/def2-tzvp(-f) level of theory, the SOCME values and transition dipole moments of T_n - S_0 phosphorescence decay at T_n geometries ($n = 2$ for compounds **1** and **2**, $n = 3$ for compound **3**) have been found to be much larger than those of T_1 - S_0 phosphorescence decay at T_1 geometries (Table 5). Phosphorescence rate constants, $k_P(T_1)$ and $k_P(T_n)$, have been obtained by FGR rate theory (Table 5). The $k_P(T_n)$ values of $10^3\sim 10^4 \text{ s}^{-1}$ at T_n geometries ($n = 2$ for compounds **1** and **2**, $n = 3$ for compound **3**) are found to be much larger than $k_P(T_1)$ values of $10^0\sim 10^1 \text{ s}^{-1}$ at T_1 geometries, exhibiting $k_P(T_n)/k_P(T_1)$ ratios of $10^2\sim 10^3$. Given that the relative emission intensity of $\text{RTP}(T_n)/\text{RTP}(T_1)$ is proportional to $k(T_1-T_n)/k(T_n-T_1) \times$

$k_P(T_n)/k_P(T_1)$, the above theoretical calculations support the experimental observation of RTP(T_n)/RTP(T_1) dual emission in the delayed spectra.

Based on the experimental observation and theoretical calculation, Figure 4a illustrates the photophysical mechanism of the dual RTP systems. Upon excitation, S_1 states of intramolecular charge transfer character form. Because of the involvement of benzophenone functional groups, different symmetry between S_1 states and triplet excited states, and relatively small singlet-triplet splitting energies, the system shows strong tendency to undergo intersystem crossing to form T_1 and T_n states. Upon intersystem crossing, the T_1 and T_n states build equilibrium under ambient conditions due to *the fast internal conversion and reverse internal conversion facilitated by electron-vibrational coupling*. The T_n states of $n-\pi^*$ characters from benzophenone groups ($n = 2$ for compounds **1** and **2**, $n = 3$ for compound **3**) have large phosphorescence decay rates and counterbalance the small population of T_n states, leading to RTP(T_n) emission that violate Kasha's rule. The PhB matrices can provide rigid microenvironment to suppress nonradiative decay (k_{nr}) and oxygen quenching (k_q) of BPBF₂'s triplet excited states. The T_1 states have small phosphorescence rates of $10^0 \sim 10^1 \text{ s}^{-1}$ to show afterglow property with lifetimes of several hundred milliseconds under ambient conditions. Because of the long-lived excited state nature, the T_1 states also serve as store for the population of T_n states via T_1 - T_n equilibrium, which is the reason that RTP(T_n) also possess lifetimes of several hundred milliseconds.

All the above results and discussion have been added in this revised manuscript.

Comment 2. Why the intensities of high-energy peak decrease upon decreasing temperature? The authors interpret it as thermally activated ISC process. I still think it come from TADF emission from S_1 due to the small bandgap between T_2 and S_1 . Why do T_2 and T_1 emission have similar phosphorescent decay times, which seem to originate from the one excited state. Therefore, I suggest the more photophysical characteristics of pure compounds of 1-3 should be collected, include temperature-dependent delayed emission spectra and decay curves.

Response 2: We thank the reviewer for the valuable comment and suggestion. The

higher-energy delayed emission band (for example, in the case of **1**-PhB-0.1%, the 421 nm delayed emission band) doesn't coincide with the steady-state emission band of **1**-PhB-0.1% at 437 nm. In addition, the significant $n-\pi^*$ transition character of T_2 states, as well as the thorough experimental studies and detailed theoretical calculations (please also find the response to Reviewer 3's Comment 1), support our assignment of the higher-energy delayed emission band to room-temperature phosphorescence from higher triplet excited states. The T_2 states are populated from the thermally activated S_1 to T_2 ISC and T_1 to T_2 reverse internal conversion, which can explain the experimental observation that the intensity of the higher-energy delayed emission bands decrease upon lowering temperature.

For the reverse ISC, it has been found that, *at the optimized geometry of either T_1 or T_n* , the T_1 and T_n levels are much lower than S_1 levels ($n = 2$ for compounds **1** and **2**, $n = 3$ for compound **3**) (Figure 1e, Figure 2d, Figure 3c and Table 2). Given that reverse ISC starts from triplet excited states, these results suggest that reverse ISC is not likely to occur. The calculated RISC rate constants of T_1 -to- S_1 and T_n -to- S_1 are small (Table S24-S29), which can explain the absence of TADF afterglow in the experimental observations.

Based on the experimental observation and theoretical calculations, we propose that, upon intersystem crossing, the T_1 and T_n states build equilibrium under ambient conditions due to the fast internal conversion and reverse internal conversion facilitated by electron-vibrational coupling. The T_n states of $n-\pi^*$ characters from benzophenone groups ($n = 2$ for compounds **1** and **2**, $n = 3$ for compound **3**) have large phosphorescence decay rates and counterbalance the small population of T_n states, leading to RTP(T_n) emission that violate Kasha's rule. The PhB matrices can provide rigid microenvironment to suppress nonradiative decay (k_{nr}) and oxygen quenching (k_q) of BPBF₂'s triplet excited states. The T_1 states of BPBF₂ compounds have small phosphorescence rates of 10^0 - 10^1 s⁻¹ to show afterglow property with lifetimes of several hundred milliseconds under ambient conditions. Because of the long-lived excited state nature, the T_1 states also serve as store for the population of T_n states via T_1 - T_n equilibrium, which is the reason that both RTP(T_n) and RTP(T_1) possess lifetimes

of several hundred milliseconds.

To study the photophysical properties of BPBF₂ in the solid state without the influence of PhB matrix, we use cyclo olefin polymer (COP) to disperse BPBF₂; COP, which has been frequently used to disperse luminescent compounds, is composed of saturated hydrocarbon and has insignificant effect on the photophysical properties of the luminescent compounds. It is found that the BPBF₂-COP samples show insignificant room-temperature afterglow. Variable temperature delayed emission spectra (1 ms delay) of BPBF₂-COP samples have been collected (Figure S4). It is found that only the delayed emission spectra at 77 K exhibit phosphorescence signals of BPBF₂ doped in COP (Figure S4). We didn't find any TADF behaviors in BPBF₂-COP samples.

From the response to Reviewer 3's Comment 1 and 2, we hope that the reviewer can accept the explanation that the higher-energy delayed emission bands originate from room-temperature phosphorescence of higher triplet excited states in the present study, rather than TADF (please also find Text S1 in the revised supporting information). Thanks very much.

Comment 3. The temperature-dependent delayed emission spectra of the PhB compound should also be performed.

Response 3: Accordingly, we collect the temperature-dependent delayed emission spectra (1 ms delay) of PhB matrix (Figure S6). It is found that the PhB matrix shows insignificant afterglow at room temperature and at low temperatures.

Comment 4. What was the critical progress compared with Previous manuscripts ((Angew. Chem. Int. Ed. 2021, 60, 17138; Adv. Funct. Mater. 2021, 2110207)?

Response 4: Our previous studies (Angew. Chem. Int. Ed. 2021, 60, 17138; Adv. Funct. Mater. 2021, 2110207) report TADF-type organic afterglow. In contrast, the present study exhibits the room-temperature afterglow from higher triplet excited states that violate Kasha's rule.

Comment 5. The authors written that “However, to the best of our knowledge, in conventional conditions, such RTP(T_n) with $\lambda_P(T_n) < \lambda_F(S_1)$ have not been observed by experimental studies.” However, as far as I know, the relative work has been reported (Angew. Chem. Int. Ed. 2020, 59, 10173 -10178).

Response 5: We thank the reviewer for this valuable information. Accordingly, we revised the corresponding sentences as follows. “However, to the best of our knowledge, in conventional conditions, such RTP(T_n) with $\lambda_P(T_n) < \lambda_F(S_1)$ have been rarely observed by experimental studies; in a reported study³⁶, RTP(T_n) signals with higher energy levels than S_1 states were collected by spectroscopic methods but the RTP(T_n) signals showed short $\tau_P < 10$ ms and cannot be observed by human eyes upon ceasing excitation source.”

Comment 6. The authors written “and the SOCME value of S_1 to T_2 transition can reach 0.91 cm^{-1} ”. The value is S_1 to T_2 or T_2 to S_1 ? Please check.

Response 6: According to the manual of ORCA, the value is T_2 to S_1 . But we would like to clarify that SOC operator used here is Hermitian, which means the SOCME of S_1 to T_2 is conjugated to the SOCME of T_2 to S_1 . The reported SOCME value is the modulus, so the SOCME value of S_1 to T_2 is the same as T_2 to S_1 .

$$\hat{H}_{so} = \frac{1}{2m_e^2c^2} \frac{1}{r} \frac{dV}{dr} \hat{L} \cdot \hat{S}$$
$$\langle S_n | \hat{H}_{so} | T_n \rangle = \langle T_n | \hat{H}_{so} | S_n \rangle^*$$

where V is the potential energy and \hat{L} and \hat{S} are the operators for orbital and spin angular momenta of electron.

Reference: Levine, I. N. *Quantum Chemistry (7th edition)*. (Pearson, 2014).

Comment 7. For compound 3, why phosphorescence intensity of T_2 is higher than T_1 ? If the 3 molecules are in monomeric form rather than in aggregation state in 3-PhB-0.1% samples, I strongly believe that high energy emission originates from TADF phenomenon rather than anti-Kasha rule.

Response 7: We thank the reviewer for the valuable comment. T_n - T_1 energy gap can be

estimated from phosphorescence maxima to be 0.38 eV, 0.33 eV and 0.26 eV for **1**-PhB-0.1% (n = 2), **2**-PhB-0.1% (n = 2) and **3**-PhB-0.1% (n = 3) powders, respectively (Table 1). The decrease of T_n - T_1 energy gap can increase the population of T_n states, which is in line with the increase of RTP(T_n)/RTP(T_1) intensity ratios in these BPBF₂-PhB-0.1% powder samples (Figure 1c, Figure 2a and Figure 3a).

In the present study, the delayed emission spectra (1 ms delay) of BPBF₂-PhB samples at different doping concentrations (0.01%, 0.1% and 1%). At low doping concentration such as 0.01%, the BPBF₂ molecules should be molecularly dispersed in PhB matrix. All these samples at different doping concentrations show the observation of higher-energy delayed emission bands have shorter emission maxima than the fluorescence bands in the steady-state emission spectra (Figure S5, S10 and S12). From all the response to Reviewer 3, we hope the reviewer can accept that the higher-energy delayed emission bands in the present study originate from room-temperature phosphorescence of higher triplet excited states rather than TADF.

We sincerely thank all the reviewers for their professional comments and constructive suggestion.

Thank you very much for your kind attention.

Yours sincerely,

Dr. Kaka Zhang

Shanghai Institute of Organic Chemistry, Chinese Academy of Sciences

Email: zhangkaka@sioc.ac.cn

Tel: +86-21-54925342; Fax: +86-21-64166128

REVIEWERS' COMMENTS

Reviewer #1 (Remarks to the Author):

The revised manuscript has addressed the previous concerns, and thus is acceptable with the condition of releasing the computational input and output files to the public domain.

Reviewer #2 (Remarks to the Author):

An uncommon system of organic-organic dopant-matrix system (BPBF2-PhB) with up-conversion RTP properties has been reported in this paper, which are characterized by the key photophysical parameters $\lambda_P(T_n, n \geq 2) < \lambda_F(S1)$ and $\tau_P > 0.1s$. After careful analysis of numerous experimental data, including some references cited by the authors, I think the authors' judgments are reasonable and defensible. In particular, the logical comparison experiments strengthened the credibility of the conclusion. The results of theoretical calculation results also support the experimental data. Although I am not an expert in theoretical calculation, the calculation programs adopted in this work are recognized by peer researchers thus the outcomes are beyond my doubt. All in all, from a personal point of view, I would like to congratulate the authors on their discovery of such rare materials and their contribution to the observation of the photophysical process in organic molecules and the system of luminous materials. In addition, after revision, the organization of the manuscript becomes more logical and the deductions are more convincing. I consider this version is acceptable to be published in Nature Communications, and the novel materials system and results will attract broad interest from readers. Just as mentioned in the paper, the anti-Kasha rule BPBF2-PhB system came from a serendipitous finding. I am interested that the participation of benzophenone functional groups on the BPBF2 molecule should not be an individual event in the realization of the up-conversion RTP in dopant-matrix systems, and that it can make $T_n (n \geq 2)$ states with analogues $3n-\pi^*$ characteristics and that large k_P values should also be generalized. I look forward to more results in the near future.

Reviewer #3 (Remarks to the Author):

Authors have adequately addressed my concerns about the original submission and revised the manuscript and SI accordingly. I thus recommend acceptance of this paper as is.

Reviewer #1 (Remarks to the Author):

The revised manuscript has addressed the previous concerns, and thus is acceptable with the condition of releasing the computational input and output files to the public domain.

Response: We sincerely thank the reviewer for the constructive comments and professional suggestion on our manuscript. The computational input and output files have been uploaded, which are available in Figshare with the identifier [<https://doi.org/10.6084/m9.figshare.22292959.v4>]. During the preparation of these files, we double-check the calculation processes and results. We find a mistake in the previous manuscript about the calculation of phosphorescence emission rate (k_P) for compound **3** (the calculations of k_P values for compound **1** and **2** are correct). This mistake is caused by the miss of the factor of 1/3 in equation 4 during calculation (equation 4 can be found in page 15 in Supplementary Information), so the calculated k_P values for compound **3** in the previous manuscript are 3 times the correct ones. I am so sorry for this mistake. We have corrected this mistake in this revised manuscript (see the yellow highlights in Table 5 and Supplementary Table 26). This mistake doesn't influence the ratio of $k_P(T_n)/k_P(T_1)$. This mistake doesn't change the order of magnitude of k_P values. I confirm that this mistake doesn't influence the discussion and conclusion in this manuscript. We hope the reviewer and editor can understand this. Thanks very much.

Reviewer #2 (Remarks to the Author):

An uncommon system of organic-organic dopant-matrix system (BPBF2-PhB) with up-conversion RTP properties has been reported in this paper, which are characterized by the key photophysical parameters $\lambda_P(T_n, n \geq 2) < \lambda_F(S1)$ and $\tau_P > 0.1s$. After careful analysis of numerous experimental data, including some references cited by the authors, I think the authors' judgments are reasonable and defensible. In particular, the logical comparison experiments strengthened the credibility of the conclusion. The results of

theoretical calculation results also support the experimental data. Although I am not an expert in theoretical calculation, the calculation programs adopted in this work are recognized by peer researchers thus the outcomes are beyond my doubt. All in all, from a personal point of view, I would like to congratulate the authors on their discovery of such rare materials and their contribution to the observation of the photophysical process in organic molecules and the system of luminous materials. In addition, after revision, the organization of the manuscript becomes more logical and the deductions are more convincing. I consider this version is acceptable to be published in Nature Communications, and the novel materials system and results will attract broad interest from readers. Just as mentioned in the paper, the anti-Kasha rule BPBF2-PhB system came from a serendipitous finding. I am interested that the participation of benzophenone functional groups on the BPBF2 molecule should not be an individual event in the realization of the up-conversion RTP in dopant-matrix systems, and that it can make T_n ($n \geq 2$) states with analogues $3n-\pi^*$ characteristics and that large k_P values should also be generalized. I look forward to more results in the near future.

Response: We sincerely thank the reviewer for the constructive comments and inspiring suggestion. We are trying more materials that may exhibit RTP(T_n) afterglow properties. Thanks very much.

Reviewer #3 (Remarks to the Author):

Authors have adequately addressed my concerns about the original submission and revised the manuscript and SI accordingly. I thus recommend acceptance of this paper as is.

Response: We sincerely thank the reviewer for the constructive comments and professional suggestion on our manuscript.

Thank you very much.

Yours sincerely,

Dr. Kaka Zhang

Shanghai Institute of Organic Chemistry, Chinese Academy of Sciences

Email: zhangkaka@sioc.ac.cn

Tel: +86-21-54925342; Fax: +86-21-64166128